# Are sample means in multi-armed bandits positively or negatively biased?

**Jaehyeok Shin**[1], **Aaditya Ramdas**[1,2] **and Alessandro Rinaldo**[1]
Department of Statistics and Data Science[1]
Machine Learning Department[2]
Carnegie Mellon University
{shinjaehyeok, aramdas, arinaldo}@cmu.edu

## Abstract

It is well known that in stochastic multi-armed bandits (MAB), the sample mean of an arm is typically not an unbiased estimator of its true mean. In this paper, we decouple three different sources of this selection bias: adaptive *sampling* of arms, adaptive *stopping* of the experiment, and adaptively *choosing* which arm to study. Through a new notion called "optimism" that captures certain natural monotonic behaviors of algorithms, we provide a clean and unified analysis of how optimistic rules affect the sign of the bias. The main takeaway message is that optimistic sampling induces a negative bias, but optimistic stopping and optimistic choosing both induce a positive bias. These results are derived in a general stochastic MAB setup that is entirely agnostic to the final aim of the experiment (regret minimization or best-arm identification or anything else). We provide examples of optimistic rules of each type, demonstrate that simulations confirm our theoretical predictions, and pose some natural but hard open problems.

## 1 Introduction

Mean estimation is one of the most fundamental problems in statistics. In the classic nonadaptive setting, we observe a fixed number of samples drawn i.i.d. from a fixed distribution with an unknown mean $\mu$. In this case, we know that the sample mean is an unbiased estimator of $\mu$.

However, in many cases the data are collected and analyzed in an adaptive manner, a prototypical example being the stochastic multi-armed bandits (MAB) framework [Robbins, 1952]. During the data collection stage, in each round an analyst can draw a sample from one among a finite set of available distributions (arms) based on the previously observed data (*adaptive sampling*). The data collecting procedure can also be terminated based on a data-driven stopping rule rather than at a fixed time (*adaptive stopping*). Further, the analyst can choose a specific target arm based on the collected data (*adaptive choosing*), for example choosing to focus on the arm with the largest empirical mean at the stopping time. In this setting, the sample mean is no longer unbiased, due to the selection bias introduced by all three kinds of adaptivity. In this paper, we provide a comprehensive understanding of the sign of the bias, decoupling the effects of these three sources of adaptivity.

In a general and unified MAB framework, we first define natural notions of monotonicity (a special case of which we call "optimism") of sampling, stopping and choosing rules. Under no assumptions on the distributions beyond assuming that their means exist, we show that optimistic sampling provably results in a negative bias, but optimistic stopping and optimistic choosing both provably result in a positive bias. Thus, the net bias can be positive or negative in general. This message is in contrast to a recent thought-provoking work by Nie et al. [2018] titled *"Why adaptively collected data has a negative bias..."* that is unfortunately misleading for practitioners, since it only analyzed the bias of adaptive sampling for a fixed arm at a fixed time.

As a concrete example, consider an offline analysis of data that was collected by an MAB algorithm (with any aim). Suppose that a practitioner wants to estimate the mean reward of some of the better arms that were picked more frequently by the algorithm. Nie et al. [2018] proved that the sample mean of each arm is negatively biased under fairly common adaptive sampling rules. Although this result is applicable only to a fixed arm at a fixed time, it could instill a possibly false sense of comfort with sample mean estimates since the practitioner might possibly think that sample means are underestimating the effect size. However, we prove that if the algorithm was adaptively stopped and the arm index was adaptively picked, then the net bias can actually be positive. Indeed, we prove that this is the case for the lil'UCB algorithm (Corollary 8), but it is likely true more generally as captured by our main theorem. Thus, the sample mean may actually overestimate the effect size. This is an important and general phenomenon for both theoreticians (to study further and quantify) and for practitioners (to pay heed to) because if a particular arm is later deployed in practice, it may yield a lower reward than was possibly expected from the offline analysis.

**Related work and our contributions.** Adaptive mean estimation, in each of the three senses described above, has received much attention in both recent and past literature. Below, we discuss how our work relates to past work, proceeding one notion at a time in approximate historical order.

We begin by noting that a single-armed bandit is simply a random walk, where adaptive stopping has been extensively studied. The book by Gut [2009] on stopped random walks is an excellent reference, summarizing almost 60 years of advances in sequential analysis. Most of these extensive results on random walks have not been extended to the MAB setting, which naturally involves adaptive sampling and choosing. Of particular relevance is the paper by Starr and Woodroofe [1968] on the sign of the bias under adaptive stopping, whose work is subsumed by ours in two ways: we not only extend their insights to the MAB setting, but even for the one-armed setting, our results generalize theirs.

Characterizing the sign of the bias of the sample mean under adaptive sampling has been a recent topic of interest due to a surge in practical applications. While estimating MAB ad revenues, Xu et al. [2013] gave an informal argument of why the sample mean is *negatively* biased for "optimistic" algorithms. Later, Villar et al. [2015] encountered this negative bias in a simulation study motivated by using MAB for clinical trials. Most recently, Bowden and Trippa [2017] derived an exact formula for the bias and Nie et al. [2018] formally provided conditions under which the bias is negative. Our results on "optimistic" sampling inducing a negative bias generalize the corresponding results in these past works.

Most importantly, however, these past results hold only at a predetermined time and for a fixed arm. Here, we put forth a complementary viewpoint that "optimistic" stopping and choosing induces a *positive* bias. Indeed, one of our central conceptual contributions is an appropriate and crisp definition of "monotonicity" and "optimism" (Definition 1), that enables a clean and general analysis.

Our main theoretical result, Theorem 7, allows the determination of the sign of the bias in several interesting settings. Importantly, the bias may be of any sign when optimistic sampling, stopping and choosing are all employed together. We demonstrate the practical validity of our theory using some simulations that yield interesting insights in their own right.

The rest of this paper is organized as follows. In Section 2, we briefly formalize the three notions of adaptivity by introducing a stochastic MAB framework. Section 3 derives results on when the bias can be positive or negative. In Section 4, we demonstrate the correctness of our theoretical predictions through simulations in a variety of practical situations. We end with a brief summary in Section 5, and for reasons of space, we defer all proofs to the Appendix.

## 2 The stochastic MAB framework

Let $P_1, \ldots, P_K$ be $K$ distributions of interest (also called arms) with finite means $\mu_k = \mathbb{E}_{Y \sim P_k}[Y]$. Every inequality and equality between two random variables is understood in the almost sure sense.

### 2.1 Formalizing the three notions of adaptivity

For those not familiar with MAB algorithms, Lattimore and Szepesvári [2019] is a good reference. The following general problem setup is critical in the rest of the paper:

- Let $W_{-1}$ denote all external sources of randomness that are independent of everything else. Draw an initial random seed $W_0 \sim U[0,1]$, and set $t = 1$.

- At time $t$, let $\mathcal{D}_{t-1}$ be the data we have so far, which is given by

$$\mathcal{D}_{t-1} := \{A_1, Y_1, \ldots, A_{t-1}, Y_{t-1}\},$$

where $A_s$ is the (random) index of arm sampled at time $s$ and $Y_s$ is the observation from the arm $A_s$. Based on the previous data (and possibly an external source of randomness), let $\nu_t(k \mid \mathcal{D}_{t-1}) \in [0,1]$ be the conditional probability of sampling the $k$-th arm for all $k \in [K] := \{1, \ldots, K\}$ with $\sum_{k=1}^{K} \nu_t(k \mid \mathcal{D}_{t-1}) = 1$. Different choices for $\nu_t$ capture commonly used methods such as random allocation, $\epsilon$-greedy [Sutton and Barto, 1998], upper confidence bound algorithms [Auer et al., 2002, Audibert and Bubeck, 2009, Garivier and Cappé, 2011, Kalyanakrishnan et al., 2012, Jamieson et al., 2014] and Thompson sampling [Thompson, 1933, Agrawal and Goyal, 2012, Kaufmann et al., 2012].

- If $W_{t-1} \in \left( \sum_{j=1}^{k-1} \nu_t(j \mid \mathcal{D}_{t-1}), \sum_{j=1}^{k} \nu_t(j \mid \mathcal{D}_{t-1}) \right)$ for some $k \in [K]$, then set $A_t = k$ which is equivalent to sample $A_t$ from a multinomial distribution with probabilities $\{\nu_t(k \mid \mathcal{D}_{t-1})\}_{k=1}^{K}$. Let $Y_t$ be a fresh independent draw from distribution $P_k$. This yields a natural filtration $\{\mathcal{F}_t\}$ which is defined, starting with $\mathcal{F}_0 = \sigma(W_{-1}, W_0)$, as

$$\mathcal{F}_t := \sigma(W_{-1}, W_0, Y_1, W_1, \ldots, Y_t, W_t), \quad \forall t \geq 1.$$

Then, $\{Y_t\}$ is adapted to $\{\mathcal{F}_t\}$, and $\{A_t\}, \{\nu_t\}$ are predictable with respect to $\{\mathcal{F}_t\}$.

- For each $k \in [K]$ and $t \geq 1$, define the running sum and number of draws for arm $k$ as $S_k(t) := \sum_{s=1}^{t} \mathbb{1}(A_s = k) Y_s$, $N_k(t) := \sum_{s=1}^{t} \mathbb{1}(A_s = k)$. Assuming that arm $k$ is sampled at least once, we define the sample mean for arm $k$ as

$$\widehat{\mu}_k(t) := \frac{S_k(t)}{N_k(t)}.$$

Then, $\{S_t\}, \{\widehat{\mu}_k(t)\}$ are adapted to $\{\mathcal{F}_t\}$ and $\{N_k(t)\}$ is predictable with respect to $\{\mathcal{F}_t\}$.

- Let $\mathcal{T}$ be a stopping time with respect to $\{\mathcal{F}_t\}$. If $\mathcal{T}$ is nonadaptively chosen, it is denoted $T$. If $t < \mathcal{T}$, draw a random seed $W_t \sim U[0,1]$ for the next round, and increment $t$. Else return the collected data $\mathcal{D}_{\mathcal{T}} = \{A_1, Y_1, \ldots, A_{\mathcal{T}}, Y_{\mathcal{T}}\} \in \mathcal{F}_{\mathcal{T}}$.

- After stopping, choose a data-dependent arm based on a possibly randomized rule $\kappa : \mathcal{D}_{\mathcal{T}} \cup \{W_{-1}\} \mapsto [K]$, but we denote the index $\kappa(\mathcal{D}_{\mathcal{T}} \cup \{W_{-1}\})$ as just $\kappa$ for short, so that the target of estimation is $\mu_\kappa$. Note that $\kappa \in \mathcal{F}_{\mathcal{T}}$, but when $\kappa$ is nonadaptively chosen (is independent of $\mathcal{F}_{\mathcal{T}}$), we called it a fixed arm and denote it as $k$.

The phrase "fully adaptive setting" refers to the scenario of running an adaptive sampling algorithm until an adaptive stopping time $\mathcal{T}$, and asking about the sample mean of an adaptively chosen arm $\kappa$. When we are not in the fully adaptive setting, we explicitly mention what aspects are adaptive.

## 2.2 The tabular perspective on stochastic MABs

It will be useful to imagine the above fully adaptive MAB experiment using a $\mathbb{N} \times K$ table, $X_\infty^*$, whose rows index time and columns index arms. Here, we put an asterisk to clarify that it is counterfactual and not necessarily observable. We imagine this entire table to be populated even before the MAB experiments starts, where for every $i \in \mathbb{N}, k \in [K]$, the $(i,k)$-th entry of the table contains an independent draw from $P_k$ called $X_{i,k}^*$. At each step, our observation $Y_t$ corresponds to the element $X_{N_k(t), A_t}^*$. Finally, we denote $\mathcal{D}_\infty^* = X_\infty^* \cup \{W_{-1}, W_0, \ldots, W_t, \ldots\}$.

Given the above tabular MAB setup (which is statistically indistinguishable from the setup described in the previous subsection), one may then find deterministic functions $f_{t,k}$ and $f_k^*$ such that

$$N_k(\mathcal{T}) = \sum_{t \geq 1} \underbrace{\mathbb{1}(A_t = k) \mathbb{1}(\mathcal{T} \geq t)}_{\mathcal{F}_{t-1}\text{-measurable}} = \sum_{t \geq 1} f_{t,k}(\mathcal{D}_{t-1}) \equiv f_k^*(\mathcal{D}_\infty^*). \tag{1}$$

Specifically, the function $f_{t,k}(\cdot)$ evaluates to one if and only if we do not stop at time $t-1$, and pull arm $k$ at time $t$. Indeed, given $\mathcal{D}_\infty^*$, the stopping time $\mathcal{T}$ is deterministic and so is the number of times

$N_k(\mathcal{T})$ that a fixed arm $k$ is pulled, and this is what $f_k^*$ captures. Along the same lines, the number of draws from a chosen arm $\kappa$ at stopping time $\mathcal{T}$ can be written in terms of the tabular data as

$$N_\kappa(\mathcal{T}) = \sum_{k=1}^{K} \mathbb{1}\left(\kappa = k\right) N_k(\mathcal{T}) \equiv \sum_{k=1}^{k} g_k^*(\mathcal{D}_\infty^*) f_k^*(\mathcal{D}_\infty^*) \tag{2}$$

for some deterministic set of functions $\{g_k^*\}$. Indeed, $g_k^*$ evaluates to one if after stopping, we choose arm $k$, which is a fully deterministic choice given $\mathcal{D}_\infty^*$.

## 3 The sign of the bias under adaptive sampling, stopping and choosing

### 3.1 Examples of positive bias due to "optimistic" stopping or choosing

In MAB problems, collecting higher rewards is a common objective of adaptive sampling strategies, and hence they are often designed to sample more frequently from a distribution which has larger sample mean than the others. Nie et al. [2018] proved that the bias of the sample mean for any *fixed* arm and at any *fixed* time is negative when the sampling strategy satisfies two conditions called "Exploit" and "Independence of Irrelevant Options" (IIO). However, the emphasis on *fixed* is important: their conditions are not enough to determine the sign of the bias under adaptive stopping or choosing, even in the simple nonadaptive sampling setting. Before formally defining our crucial notions of "optimism" in the next subsection, it is instructive to look at some examples.

**Example 1.** *Suppose we continuously alternate between drawing a sample from each of two Bernoulli distributions with mean parameters $\mu_1, \mu_2 \in (0, 1)$. This sampling strategy is fully deterministic, and thus it satisfies the Exploit and IIO conditions in Nie et al. [2018]. For any fixed time $t$, the bias equals zero for both sample means. Define a stopping time $\mathcal{T}$ as the first time we observe $+1$ from the first arm. Then the sample size of the first arm, $N_1(\mathcal{T})$, follows a geometric distribution with parameter $\mu_1$, which implies that the bias of $\widehat{\mu}_1(\mathcal{T})$ is*

$$\mathbb{E}\left[\widehat{\mu}_1(\mathcal{T}) - \mu_1\right] = \mathbb{E}\left[\frac{1}{N_1(\mathcal{T})}\right] - \mu_1 = \frac{\mu_1 \log(1/\mu_1)}{1 - \mu_1} - \mu_1,$$

*which is positive for all $\mu_1 \in (0, 1)$.*

This example shows that for nonadaptive sampling, adaptive stopping can induce a *positive* bias. In fact, this example is not atypical, but is an instance of a more general phenomenon explored in the one-armed setting in sequential analysis. For example, Siegmund [1978, Ch. 3] contains the following classical result for a Brownian motion $W(t)$ with positive drift $\mu > 0$.

**Example 2.** *If we define a stopping time as the first time $W(t)$ exceeds a line with slope $\eta$ and intercept $b > 0$, that is $\mathcal{T}_B := \inf\{t \geq 0 : W(t) \geq \eta t + b\}$, then for any slope $\eta \leq \mu$, we have $\mathbb{E}\left[\frac{W(\mathcal{T}_B)}{\mathcal{T}_B} - \mu\right] = 1/b$. Note that a sum of Gaussians with mean $\mu$ behaves like a time-discretization of a Brownian motion with drift $\mu$; since $\mathbb{E}W(t) = t\mu$, we may interpret $W(\mathcal{T}_B)/\mathcal{T}_B$ as a stopped sample mean, and the last equation implies that its bias is $1/b$, which is positive.*

Generalizing further, Starr and Woodroofe [1968] proved the following remarkable result.

**Example 3.** *If we stop when the sample mean crosses any predetermined upper boundary, the stopped sample mean is always positive biased (whenever the stopping time is a.s. finite). Explicitly, choosing any arbitrary sequence of real-valued constants $\{c_k\}$, define $\mathcal{T}_c := \inf\{t : \widehat{\mu}_1(t) > c_t\}$, then as long as the observations $X_i$ have a finite mean and $\mathcal{T}_c$ is a.s. finite, we have $\mathbb{E}\left[\widehat{\mu}_1(\mathcal{T}_c) - \mu_1\right] - \mu_1 > 0$.*

Surprisingly, we will generalize the above strong result even further. Additionally, stopping times in the MAB literature can be thought of as extensions of $\mathcal{T}_c$ and $\mathcal{T}_B$ to a setting with multiple arms, and we will prove that indeed the bias induced will still be positive. We end with an example of the positive bias induced by "optimistic" choosing:

**Example 4.** *Given $K$ standard normals $\{Z_i\}$ (to be thought of as one sample from each of $K$ arms), let $\kappa = \operatorname{argmax}_k Z_k$, that is, we choose the arm with the largest observation. It is well known that $\mathbb{E}\left[Z_\kappa\right] = \mathbb{E}\left[\max_{k \in [K]} Z_k\right] \asymp \sqrt{2 \log K}$. Since $\mathbb{E}Z_k = 0$ for all $k$, but $\mathbb{E}Z_\kappa > 0$, the "optimistic" choice $\kappa$ induces a positive bias.*

In many typical MAB settings, we should expect sample means to have two contradictory sources of bias: negative bias from "optimistic sampling" and positive bias from "optimistic stopping/choosing".

## 3.2 Positive or negative bias under monotonic sampling, stopping and choosing

Based on the expression (2), we formally state a characteristic of data collecting strategies which fully determines the sign of the bias as follows.

**Definition 1.** *A data collecting strategy is "monotonically increasing (or decreasing)" if for any $i \in \mathbb{N}$ and $k \in [K]$, the function $\mathcal{D}_\infty^* \mapsto g_k^*(\mathcal{D}_\infty^*)/f_k^*(\mathcal{D}_\infty^*) \equiv \mathbb{1}(\kappa = k)/N_k(\mathcal{T})$, is an increasing (or decreasing) function of $X_{i,k}^*$ while keeping all other entries in $\mathcal{D}_\infty^*$ fixed. Further, we say that*

- *a data collecting strategy has an optimistic sampling rule if the function $\mathcal{D}_\infty^* \mapsto N_k(t)$ is an increasing function of $X_{i,k}^*$ while keeping all other entries in $\mathcal{D}_\infty^*$ fixed for any fixed $i \in \mathbb{N}$, $t \geq 1$ and $k \in [K]$;*

- *a data collecting strategy has an optimistic stopping rule if $\mathcal{D}_\infty^* \mapsto \mathcal{T}$ is a decreasing function of $X_{i,k}^*$ while keeping all other entries in $\mathcal{D}_\infty^*$ fixed for any fixed $i \in \mathbb{N}$ and $k \in [K]$;*

- *a data collecting strategy has an optimistic choosing rule if $\mathcal{D}_\infty^* \mapsto \mathbb{1}(\kappa = k)$ is an increasing function of $X_{i,k}^*$ while keeping all other entries in $\mathcal{D}_\infty^*$ fixed for any fixed $i \in \mathbb{N}$ and $k \in [K]$.*

Note that if a data collecting strategy has an optimistic sampling (or stopping or choosing) rule, with the other components being nonadaptive, then the strategy is monotonically decreasing (increasing). We remark that nonadaptive just means independent of the entries $X_{i,k}^*$, but it is not necessarily deterministic[1]. The above definition warrants some discussion to provide intuition.

Roughly speaking, under optimistic stopping, if a sample from the $k$-th distribution was increased while keeping all other values fixed, the algorithm would reach its termination criterion sooner. For instance, $\mathcal{T}_B$ from Example 2 and the criterion in Example 1 are both optimistic stopping rules. Most importantly, boundary-crossing is optimistic:

**Fact 1.** *The general boundary-crossing stopping rule of Starr and Woodroofe [1968], denoted $\mathcal{T}_c$ in Example 3, is an optimistic stopping rule (and hence optimistic stopping is a weaker condition).*

Optimistic stopping rules do not need to be based on the sample mean; for example, if $\{c_t\}$ is an arbitrary sequence, then $\mathcal{T} := \inf\{t \geq 3 : X_t + X_{t-2} \geq c_t\}$ is an optimistic stopping rule. In fact, $\mathcal{T}_\ell := \inf\{t \geq 3 : \ell_t(X_1, \ldots, X_t) \geq c_t\}$ is optimistic, as long as each $\ell_t$ is coordinatewise nondecreasing.

For optimistic choosing, the previously discussed argmax rule (Example 4) is optimistic. More generally, it is easy to verify the following:

**Fact 2.** *For any probabilities $p_1 \geq p_2 \cdots \geq p_K$ that sum to one, a rule that chooses the arm with the $k$-th largest empirical mean with probability $p_k$, is an optimistic choosing rule.*

Turning to the intuition for optimistic sampling, if a sample from the $k$-th distribution was increased while keeping all other values fixed, the algorithm would sample the $k$-th arm more often. We claim that optimistic sampling is a weaker condition than the Exploit and IIO conditions employed by Nie et al. [2018].

**Fact 3.** *The "Exploit" and "IIO" conditions in Nie et al. [2018] together imply that the sampling strategy is optimistic (and hence optimistic sampling is a weaker condition). Further, as summarized in Appendix A, $\epsilon$-greedy, UCB and Thompson sampling (Gaussian-Gaussian and Beta-Bernoulli, for instance) are all optimistic sampling methods.*

For completeness, we prove the first part formally in Appendix A.2, which builds heavily on observations already made in the proof of Theorem 1 in Nie et al. [2018]. Beyond the instances mentioned above, Corollary 10 in the supplement captures a sufficient condition for Thompson sampling with one-dimensional exponential families and conjugate priors to be optimistic. We now provide an expression for the bias that holds at any stopping time and for any sampling algorithm.

**Proposition 5.** *Let $\mathcal{T}$ be a stopping time with respect to the natural filtration $\{\mathcal{F}_t\}$. For each fixed $k \in [K]$ such that $0 < \mathbb{E}N_k(\mathcal{T}) < \infty$, the bias of $\widehat{\mu}_k(\mathcal{T})$ is given as*

$$\mathbb{E}\left[\widehat{\mu}_k(\mathcal{T}) - \mu_k\right] = -\frac{\mathrm{Cov}\left(\widehat{\mu}_k(\mathcal{T}), N_k(\mathcal{T})\right)}{\mathbb{E}\left[N_k(\mathcal{T})\right]}. \tag{3}$$

The proof may be found in Appendix B.3. A similar expression was derived in Bowden and Trippa [2017], but only for a fixed time $T$. In order to extend it to stopping times (that are allowed to be infinite, as long as $\mathbb{E}N_k(\mathcal{T}) < \infty$), we derive a simple generalization of Wald's first identity to the MAB setting. Specifically, recalling that $S_k(t) = \widehat{\mu}_k(t)N_k(t)$, we show the following:

**Lemma 6.** *Let $\mathcal{T}$ be a stopping time with respect to the natural filtration $\{\mathcal{F}_t\}$. For each fixed $k \in [K]$ such that $\mathbb{E}N_k(\mathcal{T}) < \infty$, we have $\mathbb{E}[S_k(\mathcal{T})] = \mu_k\mathbb{E}[N_k(\mathcal{T})]$.*

This lemma is also proved in Appendix B.3. Proposition 5 provides a simple, and somewhat intuitive, expression of the bias for each arm. It implies that if the covariance of the sample mean of an arm and the number of times it was sampled is positive (negative), then the bias is negative (positive). We now formalize this intuition below, including for adaptively chosen arms. The following theorem shows that if the adaptive sampling, stopping and choosing rules are monotonically increasing (or decreasing), then the sample mean is positively (or negatively) biased.

**Theorem 7.** *Let $\mathcal{T}$ be a stopping time with respect to the natural filtration $\{\mathcal{F}_t\}$ and let $\kappa : \mathcal{D}_\mathcal{T} \mapsto [K]$ be a choosing rule. Suppose each arm has finite expectation and, for all $k$ with $\mathbb{P}(\kappa = k) > 0$, we have $\mathbb{E}[N_k(\mathcal{T})] < \infty$ and $N_k(\mathcal{T}) \geq 1$. If the data collecting strategy is monotonically decreasing, for example under optimistic sampling with nonadaptive stopping and choosing, then we have*

$$\mathbb{E}\left[\widehat{\mu}_\kappa(\mathcal{T}) \mid \kappa = k\right] \leq \mu_k, \ \ \forall k : \mathbb{P}(\kappa = k) > 0, \tag{4}$$

*which also implies that*

$$\mathbb{E}\left[\widehat{\mu}_\kappa(\mathcal{T}) - \mu_\kappa\right] \leq 0. \tag{5}$$

*Similarly if the data collecting strategy is monotonically increasing, for example under optimistic stopping with nonadaptive sampling and choosing, or under optimistic choosing with nonadaptive sampling and stopping, then we have*

$$\mathbb{E}\left[\widehat{\mu}_\kappa(\mathcal{T}) \mid \kappa = k\right] \geq \mu_k, \ \ \forall k : \mathbb{P}(\kappa = k) > 0, \tag{6}$$

*which also implies that*

$$\mathbb{E}\left[\widehat{\mu}_\kappa(\mathcal{T}) - \mu_\kappa\right] \geq 0. \tag{7}$$

*If each arm has a bounded distribution then the condition $\mathbb{E}[N_k(\mathcal{T})] < \infty$ can be dropped.*

**Remark 1.** *In fact, if each arm has a finite $p$-th moment for a fixed $p > 2$ then the condition $\mathbb{E}[N_k(\mathcal{T})] < \infty$ can be dropped.*

The proofs of Theorem 7 and Remark 1 can be found in Appendix B.1 and are based on martingale arguments that are quite different from the ones used in Nie et al. [2018]. See also Appendix A.4 for an intuitive explanation of the sign of the bias under optimistic sampling, stopping or choosing rules. The expression (3) intuitively suggests situations when the sample mean estimator $\widehat{\mu}_k(\mathcal{T})$ is biased, while the inequalities in (4) and (6) determine the direction of bias under the monotonic or optimistic conditions. Due to Facts 1, 2 and 3, several existing results are immediately subsumed and generalized by Theorem 7. Further, the following corollary is a particularly interesting special case dealing with the lil'UCB algorithm by Jamieson et al. [2014] which uses adaptive sampling, stopping and choosing, as summarized in Section 4.3.

**Corollary 8.** *The lil'UCB algorithm is a monotonically increasing strategy, and thus the sample mean of the reported arm when lil'UCB stops is always positively biased.*

The proof is described in Appendix B.2. The above result is interesting because of the following reasons: (a) when viewed separately, the sampling, stopping and choosing rules of the lil'UCB algorithm all seem to be optimistic (however, they are not optimistic, because our definition requires two out of three to be nonadaptive); hence it is apriori unclear which rule dominates and whether the net bias should be positive or negative; (b) we did not have to alter anything about the algorithm in order to prove that it is a monotonically increasing strategy (for any distribution over arms, for any number of arms). The generality of the above result showcases the practical utility of our theorem, whose message is in sharp contrast to the title of the paper by Nie et al. [2018].

Next, we provide simulation results that verify that our monotonic and optimistic conditions accurately capture the sign of the bias of the sample mean.

# 4 Numerical experiments

## 4.1 Negative bias from optimistic sampling rules in multi-armed bandits

Recall Fact 3, which stated that common MAB adaptive sampling strategies like greedy (or $\epsilon$-greedy), upper confidence bound (UCB) and Thompson sampling are optimistic. Thus, for a deterministic stopping time, Theorem 7 implies that the sample mean of each arm is always negatively biased. To demonstrate this, we conduct a simulation study in which we have three unit-variance Gaussian arms with $\mu_1 = 1, \mu_2 = 2$ and $\mu_3 = 3$. After sampling once from each arm, greedy, UCB and Thompson sampling are used to continue sampling until $T = 200$. We repeat the whole process from scratch $10^4$ times for each algorithm to get an accurate estimate for the bias.[2] Due to limited space, we present results from UCB and Thompson sampling only but detailed configurations of algorithms and a similar result for the greedy algorithm can be found in Appendix C.1. Figure 1 shows the distribution of observed differences between sample means and the true mean for each arm. Vertical lines correspond to biases. The example demonstrates that the sample mean is negatively biased under optimistic sampling rules.

**Remark 2.** *The main goal in our simulations is to visualize and corroborate our theoretical results about the sign of the bias. As a result, we do not make any attempt to optimize the parameters for UCB or Thompon sampling for the purpose of minimizing the regret, since the latter is not the paper's aim. However, investigating the relationship between the performance of MAB algorithms and the bias at the time horizon would be an interesting future direction of research.*

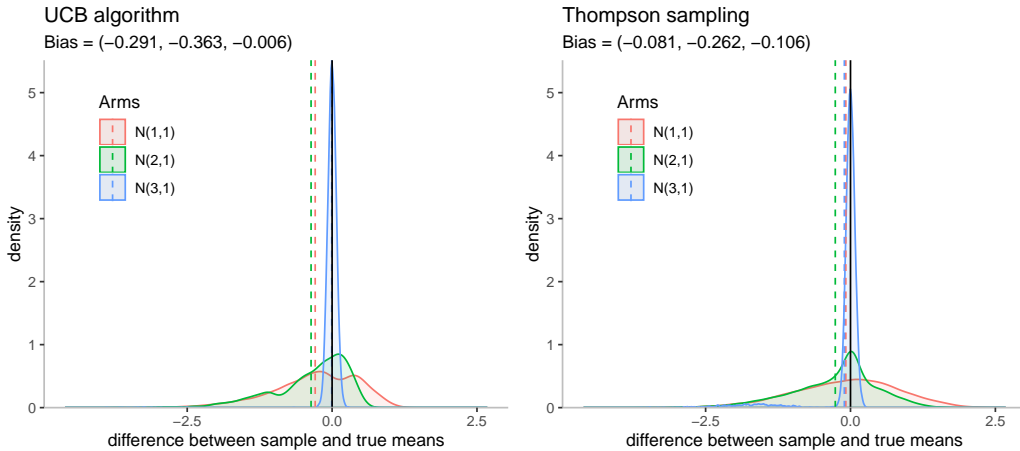

Figure 1: *Data is collected by UCB (left) and Thompson sampling (right) algorithms from three unit-variance Gaussian arms with $\mu_1 = 1, \mu_2 = 2$ and $\mu_3 = 3$. For all three arms, sample means are negatively biased (at fixed times). A similar result for the greedy algorithm can be found in Appendix C.1.*

## 4.2 Bias from stopping a one-sided sequential likelihood ratio test

Suppose we have two independent sub-Gaussian arms with common and known parameter $\sigma^2$ but unknown means $\mu_1$ and $\mu_2$. Consider the following testing problem:

$$H_0 : \mu_1 \leq \mu_2 \quad \text{vs} \quad H_1 : \mu_1 > \mu_2.$$

To test this hypothesis, suppose we draw a sample from arm 1 for every odd time and from arm 2 for every even time. Instead of conducting a test at a fixed time, we can use the following one-sided sequential likelihood ratio test [Robbins, 1970, Howard et al., 2018]: for any fixed $w > 0$ and $\alpha \in (0,1)$, define a stopping time $\mathcal{T}$ as

$$\mathcal{T}^w := \inf \left\{ t \in \mathbb{N}_{\text{even}} : \widehat{\mu}_1(t) - \widehat{\mu}_2(t) \geq \frac{2\sigma}{t} \sqrt{(t + 2w) \log \left( \frac{1}{2\alpha} \sqrt{\frac{t + 2w}{2w}} + 1 \right)} \right\}, \quad (8)$$

where $\mathbb{N}_{\text{even}} := \{2n : n \in \mathbb{N}\}$. For a given fixed maximum even time $M \geq 2$, we stop sampling at time $\mathcal{T}_M^w := \min\{\mathcal{T}^w, M\}$. Then, we reject the null $H_0$ if $\mathcal{T}_M^w < M$. It can be checked [Howard et al., 2018, Section 8] that, for any fixed $w > 0$, this test controls the type-1 error at level $\alpha$ and the power goes to 1 as $M$ goes to infinity.

For the arms 1 and 2, these are special cases of optimistic and pessimistic stopping rules respectively. From Theorem 7, we have that $\mu_1 \leq \mathbb{E}\widehat{\mu}_1(\mathcal{T}_M^w)$ and $\mu_2 \geq \mathbb{E}\widehat{\mu}_2(\mathcal{T}_M^w)$. To demonstrate this, we conduct two simulation studies with unit variance Gaussian errors: one under the null hypothesis $(\mu_1, \mu_2) = (0, 0)$, and one under the alternative hypothesis $(\mu_1, \mu_2) = (1, 0)$. We choose $M = 200$, $w = 10$ and $\alpha = 0.1$. As before, we repeat each experiment $10^4$ times for each setting. Figure 2 shows the distribution of observed differences between sample means and the true mean for each arm under null and alternative hypothesis cases. Vertical lines correspond to biases. The simulation study demonstrates that the sample mean for arm 1 is positively biased and the sample mean for arm 2 is negatively biased as predicted.

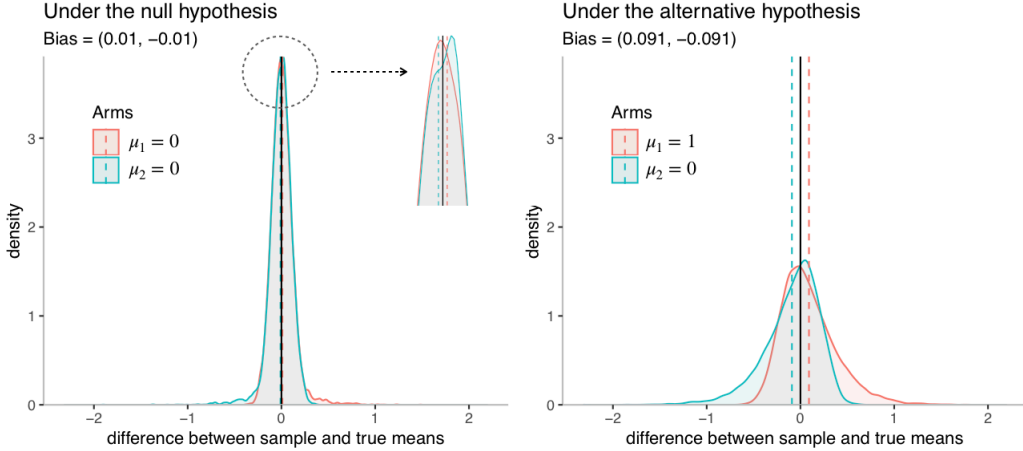

Figure 2: *Data is collected from the one-sided sequential likelihood ratio test procedure described in Section 4.2. The sample mean for arm 1 is positively biased and the sample mean for arm 2 is negatively biased under both null and alternative hypothesis cases. Note that the size of the bias under the null hypothesis is smaller than the one under the alternative hypothesis since the number of collected samples is larger under the null hypothesis.*

### 4.3 Positive bias of the lil'UCB algorithm in best-arm identification

Suppose we have $K$ sub-Gaussian arms with mean $\mu_1, \ldots, \mu_K$ and known parameter $\sigma$. In the best-arm identification problem, our target of inference is the arm with the largest mean. There exist many algorithms for this task including lil'UCB [Jamieson et al., 2014], Top-Two Thompson Sampling [Russo, 2016] and Track-and-Stop [Garivier and Kaufmann, 2016].

In Corollary 8, we showed that the lil'UCB algorithm is monotonically increasing, and thus the sample mean of the chosen arm is positively biased. In this subsection, we verify it with a simulation. It is an interesting open question whether different types of best-arm identification algorithms also yield positively biased sample means.

The lil'UCB algorithm consists of the following optimistic sampling, stopping and choosing:

- Sampling: For any $k \in [K]$ and $t = 1, \ldots K$, define $\nu_t(k) = \mathbb{1}(t = k)$. For $t > K$,

$$\nu_t(k) = \begin{cases} 1 & \text{if } k = \operatorname{argmax}_{j \in [K]} \widehat{\mu}_j(t-1) + u_t^{\text{lil}}\left(N_j(t-1)\right), \\ 0 & \text{otherwise}, \end{cases}$$

  where $\delta, \epsilon, \lambda$ and $\beta$ are algorithm parameters and

$$u_t^{\text{lil}}(n) := (1 + \beta)(1 + \sqrt{\epsilon})\sqrt{2\sigma^2(1 + \epsilon)\log\left(\log((1 + \epsilon)n)/\delta\right)/n}.$$

- Stopping: $\mathcal{T} = \inf\left\{t > K : N_k(t) \geq 1 + \lambda \sum_{j \neq k} N_j(t) \text{ for some } k \in [K]\right\}$.

- Choosing: $\kappa = \mathrm{argmax}_{k \in [K]} N_k(\mathcal{T})$.

Once we stop sampling at time $\mathcal{T}$, the lil'UCB algorithm guarantees that $\kappa$ is the index of the arm with largest mean with some probability depending on input parameters. Based on this, we can also estimate the largest mean by the chosen stopped sample mean $\widehat{\mu}_\kappa(\mathcal{T})$. The performance of this sequential procedure can vary based on underlying distribution of the arm and the choice of parameters. However, we can check this optimistic sampling and optimistic stopping/choosing rules which would yield negative and positive biases respectively are monotonic increasing and thus the chosen stopped sample mean $\widehat{\mu}_\kappa(\mathcal{T})$ is always positively biased for any choice of parameters.

To verify it with a simulation, we set 3 unit-variance Gaussian arms with means $(\mu_1, \mu_2, \mu_3) = (g, 0, -g)$ for each gap parameter $g = 1, 3, 5$. We conduct $10^4$ trials of the lil'UCB algorithm with a valid choice of parameters described in Jamieson et al. [2014, Section 5]. Figure 3 shows the distribution of observed differences between the chosen sample means and the corresponding true mean for each $\delta$. Vertical lines correspond to biases. The simulation study demonstrates that, in all configurations, the chosen stopped sample mean $\widehat{\mu}_\kappa(\mathcal{T})$ is always positively biased. (see Appendix B.2 for a formal proof.)

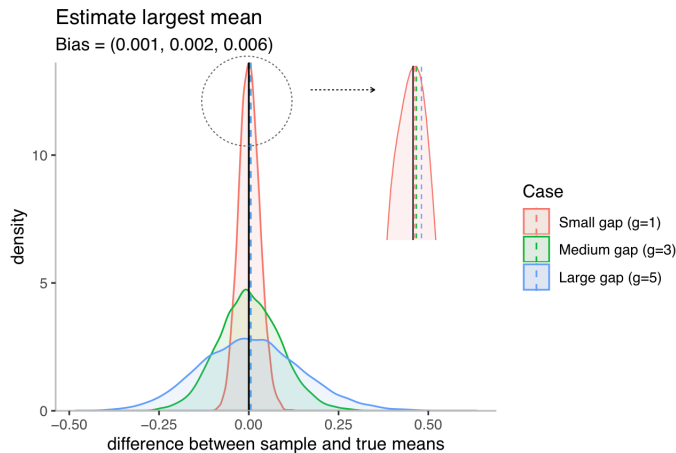

Figure 3: *Data is collected by the lil'UCB algorithm run on three unit-variance Gaussian arms with $\mu_1 = g, \mu_2 = 0$ and $\mu_3 = -g$ for each gap parameter $g = 1, 3, 5$. For all cases, chosen sample means are positively biased. The bias is larger for a larger gap since the number of collected samples is smaller on an easier task.*

## 5    Summary

This paper provides a general and comprehensive characterization of the sign of the bias of the sample mean in multi-armed bandits. Our main conceptual innovation was to define new weaker conditions (monotonicity and optimism) that capture a wide variety of practical settings in both the random walk (one-armed bandit) setting and the MAB setting. Using this, our main theoretical contribution, Theorem 7, significantly generalizes the kinds of algorithms or rules for which we can mathematically determine the sign of the bias for any problem instance. Our simulations confirm the accuracy of our theoretical predictions for a variety of practical situations for which such sign characterizations were previously unknown. There are several natural followup directions: (a) extending results like Corollary 8 to other bandit algorithms, (b) extending all our results to hold for other functionals of the data like the sample variance, (c) characterizing the magnitude of the bias. We have recently made significant progress on the last question [Shin et al., 2019], but the other two remain open.

## Footnotes

[1]An example of a random but nonadaptive stopping rule: flip a (potentially biased) coin at each step to decide whether to stop. An example of a random but nonadaptive sampling rule: with probability half pick a uniformly random arm, and with probability half pick the arm that has been sampled most often thus far.

[2]In all experiments, sizes of reported biases are larger than at least 3 times the Monte Carlo standard error.

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
