[Supplementary Material 1 · bias_NeurIPS_2019_camera_ready_appendix.pdf]

# A    $\epsilon$-greedy, UCB and Thompson sampling are optimistic sampling rules

## A.1    Descriptions of $\epsilon$-greedy, UCB and Thompson sampling rules

$\epsilon$-greedy, UCB, and Thomson sampling have the following sampling rules.

- $\epsilon$-greedy algorithm : For any $k \in [K]$ and $t \in [\mathcal{T}]$,

$$\nu_t(k) = \begin{cases} 1 - \epsilon & \text{if } k = \text{argmax}_{j \in [K]} \, \widehat{\mu}_j(t-1), \\ \frac{\epsilon}{K-1} & \text{otherwise.} \end{cases}$$

- UCB : For any $k \in [K]$ and $t \in [\mathcal{T}]$,

$$\nu_t(k) = \begin{cases} 1 & \text{if } k = \text{argmax}_{j \in [K]} \, \widehat{\mu}_j(t-1) + u_{t-1}(S_j(t-1), N_j(t-1)), \\ 0 & \text{otherwise,} \end{cases}$$

  where $(s, n) \mapsto u_{t-1}(s, n)$ is a non-negative function which is increasing and decreasing with respect to the first and second inputs respectively for each $t$. For example, a simple version of UCB uses $u_{t-1}(s, n) = \sqrt{\frac{2 \log(1/\delta)}{n}}$ for a properly chosen constant $\delta \in (0, 1)$.

- Thompson sampling : For any $k \in [K]$ and $t \in [\mathcal{T}]$,

$$\nu_t(k) \propto \pi(k = \underset{j}{\text{argmax}}\, \mu_j \mid A_1, Y_1, \dots, A_{t-1}, Y_{t-1}).$$

  where $\pi$ is a prior on $(\mu_1, \dots, \mu_K)$ or, more generally, on parameters of arms $(\theta_1, \dots, \theta_K)$. In particular, if underlying arms are Gaussian with common variance $\sigma^2$ and if we impose independent Gaussian prior $N(\mu_{k,0}, \sigma_0^2)$ on each arm $k$, the corresponding Thompson sampling is statistically equivalent to the following rule.

$$\nu_t(k) = \begin{cases} 1 & \text{if } k = \text{argmax}_{j \in [K]} \, \widetilde{\mu}_j(t-1) + \sigma_j(t-1) Z_{j,t-1} \\ 0 & \text{otherwise,} \end{cases}$$

  where each $Z_{j,t-1}$ is an independent draw from $N(0, 1)$ and $\widetilde{\mu}_j(t-1), \sigma_k(t-1)$ are the posterior mean and standard deviation of arm $j$, given as

$$\widetilde{\mu}_j(t-1) = \frac{\mu_{j,0}/\sigma_0^2 + N_j(t-1)\widehat{\mu}_j(t-1)/\sigma^2}{1/\sigma_0^2 + N_j(t-1)/\sigma^2}, \quad \sigma_j(t-1) = \left(1/\sigma_0^2 + N_j(t-1)/\sigma^2\right)^{-1/2}.$$

## A.2    Exploit and IIO conditions are sufficient for optimistic sampling

In Fact 3, we claimed that "Exploit" and "IIO" conditions in Nie et al. [2018] are jointly a sufficient condition for a sampling rule being optimistic. In this subsection, we formally restate Exploit and IIO conditions of Nie et al. [2018] in terms of our notations and prove Fact 3.

First, fix a deterministic stopping time $T$. Given any $t \in [T], k \in [K]$, define respectively the data from arm $k$ until time $t$, and the data from all arms except $k$ until time $t$, as

$$\mathcal{D}_t^{(k)} := \left\{X_{i,k}^*\right\}_{i=1}^{N_k(t)} \quad \text{and} \quad \mathcal{D}_t^{(-k)} := \mathcal{D}_t \setminus \mathcal{D}_t^{(k)} = \bigcup_{j \neq k} \left\{X_{i,j}^*\right\}_{i=1}^{N_j(t)} \cup \{W_{-1}, W_0, \dots, W_t\},$$

where $\mathcal{D}_t$ is the sample history up to time $t$ under a tabular model $\mathcal{D}_\infty^*$. Let $\mathcal{D}_\infty^{*'}$ be another tabular model. Under $\mathcal{D}_\infty^{*'}$, we define $\mathcal{D}_t', \mathcal{D'}_t^{(k)}$ and $\mathcal{D'}_t^{(-k)}$ in the same way. The Exploit condition in Nie et al. [2018] can be rewritten as following.

**Definition 2** (Exploit). *Given any $t \in [T], k \in [K]$, suppose $\mathcal{D}_t^{(k)}$ and $\mathcal{D'}_t^{(k)}$ have the same size (that is $N_k'(t) = N_k(t)$) and $\mathcal{D}_t^{(-k)} = \mathcal{D'}_t^{(-k)}$. If the sample mean $\widehat{\mu}_k(t)$ under $\mathcal{D}_t^{(k)}$ is less than or equal to the sample mean $\widehat{\mu}_k'(t)$ under $\mathcal{D'}_t^{(k)}$, then*

$$\mathbb{1}(A_t = k) := f_{t,k}\left(\mathcal{D}_t^{(k)} \cup \mathcal{D}_t^{(-k)}\right) \le f_{t,k}\left(\mathcal{D'}_t^{(k)} \cup \mathcal{D}_t^{(-k)}\right) =: \mathbb{1}(A_t' = k).$$

For the IIO condition, we present a specific version in the MAB setting which was originally used in Eq.(8) in the proof of Theorem 1 in Nie et al. [2018].

**Definition 3** (Independence of Irrelevant Options (IIO))**.** *For each $t, k$, the sampling random variable $A_t$ can be written in terms of deterministic functions $f_{t,k}$ and $g_{t,k}$ such that*

$$A_t = \begin{cases} k & \text{if } f_{t,k}\left(\mathcal{D}_{t-1}\right) = 1 \\ j & \text{if } f_{t,k}\left(\mathcal{D}_{t-1}\right) = 0 \ \text{ and } \ g_{t,k}\left(\mathcal{D}_{t-1}^{(-k)}\right) = j \ \text{for some } j \neq k. \end{cases}$$

Intuitively, $f_{t,k}$ is simply the indicator of whether arm $k$ was pulled at time $t$; the crucial part is $g_{t,k}$, which specifies which arm is selected when arm $k$ is not, and the IIO condition requires that $g_{t,k}$ ignores the data from arm $k$ in order to determine which $j \neq k$ to pull instead.

It can be checked that $\epsilon$-greedy, UCB and Thompson sampling under Gaussian arms and Gaussian priors satisfy both conditions. Indeed, if arm $k$ is not the arm with the highest mean or highest UCB (for example), determining which other arm does get pulled in the next step does not depend on the data from arm $k$. In Appendix A.3, we present a sufficient condition for Thompson sampling to satisfy both conditions, and thus to be optimistic which shows Thompson sampling is optimistic for many commonly used exponential family arms including Gaussian, Bernoulli, exponential and Possion arms with their conjugate priors.

Before we prove Fact 3, we first introduce a lemma related to the IIO condition as follows.

**Lemma 9.** *Fix a $k \in [K]$. Let $\mathcal{D}_\infty^*$ and $\mathcal{D}_\infty^{*'}$ be two MAB tabular representation that agree with each other except in their $k$-th column. Let $N_j(t)$ and $N_j'(t)$ be the numbers of draws from arm $j$ for all $j \in [K]$ under $\mathcal{D}_\infty^*$ and $\mathcal{D}_\infty^{*'}$ respectively. Then, under IIO, the following implication holds:*

$$N_k(t) \leq N_k'(t) \Rightarrow N_j(t) \geq N_j'(t), \quad \text{for all } j \neq k. \tag{9}$$

*By switching the roles of $\mathcal{D}_\infty^*$ and $\mathcal{D}_\infty^{*'}$, we also have*

$$N_k(t) \geq N_k'(t) \Rightarrow N_j(t) \leq N_j'(t), \quad \text{for all } j \neq k, \tag{10}$$

*and therefore,*

$$N_k(t) = N_k'(t) \Rightarrow N_j(t) = N_j'(t), \quad \text{for all } j \neq k. \tag{11}$$

*Proof of Lemma 9.* It is enough to prove the first statement. We follow the logic in the proof of Property 1 in Nie et al. [2018]. If $N_k(t) = t$ or $N_k'(t) = t$ then the claimed statement holds trivially since $N_j(t) + N_k(t) \leq t$ and $N_j'(t) + N_k'(t) \leq t$ for all $j \neq k$. Therefore, for the rest of the proof, we assume $N_k(t) \leq N_k'(t) < t$.

For each $t$, define $s_1 < \cdots < s_{t-N_k(t)}$ to be the sequence of times at which arm $k$ was *not* sampled before time $t$ under $\mathcal{D}_\infty^*$. Similarly, let $s_1' < \cdots < s_{t-N_k(t)}'$ be the sequence of times at which arm $k$ was *not* sampled before time $t$ under $\mathcal{D}_\infty^{*'}$. From the IIO condition and the assumption that $\mathcal{D}_\infty^*$ and $\mathcal{D}_\infty^{*'}$ agree with each other except in their $k$-th column, we have

$$A_{s_u} = A_{s_u'}, \quad \text{for all } u \in \{1, \ldots, t - N_k'(t)\}, \tag{12}$$

which implies that

$$N_j'(t) = N_j'(s_{t-N_k'(t)}') = N_j(s_{t-N_k'(t)}) \leq N_j(s_{t-N_k(t)}) = N_j(t),$$

where the first and the last identities stem from the definition of $s$ and $s'$, the second identity is due to (12), and the inequality follows from the assumption that $N_k(t) \leq N_k'(t)$ along with the fact that $u \mapsto s_u$ and $s \mapsto N_j(s)$ are increasing. $\qquad\square$

*Proof of Fact 3.* Let us fix an arm $k$ and a deterministic stopping time $T$, and a time $t \leq T$, as required by Exploit and IIO conditions. The arguments below are inspired by case 1 in the proof of Theorem 1 in Nie et al. [2018].

Let $X_{i,k}^{*'}$ be an independent copy of $X_{i,k}^*$ and define $X_\infty^{*'}$ as a $\mathbb{N} \times K$ table which equals $X_\infty^*$ on all entries except the $(i, k)$-th entry, which contains $X_{i,k}^{*'}$. Let $\mathcal{D}_\infty^{*'} = X_\infty^{*'} \cup \{W_{-1}, W_0, \ldots\}$ denote

the corresponding dataset, which only differs from $\mathcal{D}^*_\infty$ in one element. Let $N_k(T)$ and $N'_k(T)$ be numbers of draws from arm $k$ up to time $T$ based on $\mathcal{D}^*_\infty$ and $\mathcal{D}^{*'}_\infty$ respectively. Also for each $t \le T$, let $A_t$ and $A'_t$ be sampled arms based on $\mathcal{D}^*_\infty$ and $\mathcal{D}^{*'}_\infty$ respectively.

To prove the claim, it is enough to show that if $X^*_{i,k} \le X^{*'}_{i,k}$ then $N_k(T) \le N'_k(T)$ under Exploit and IIO conditions. Suppose, for the sake of deriving a contradiction, that there exist $i \in \mathbb{N}$ and $k \in [K]$ such that $X^*_{i,k} \le X^{*'}_{i,k}$ but $N_k(T) > N'_k(T)$. Note that since $A_s$ and $A'_s$ are functions of the history up to time $s-1$, we know that $A_s = A'_s$ for all $s \le t$, where $t$ is defined as $t = \min\left\{s \ge 1 : N_k(s) = N'_k(s) = i\right\}$. If $t \ge T$, we have that $N_k(T) = N_k(t) = N'_k(t) = N'_k(T)$, which contradicts our assumption. Hence, we may assume $t < T$ for the rest of the proof.

Define $s_0 := \min\left\{s \ge 1 : N_k(s) > N'_k(s)\right\}$. From the definition of $s_0$, we know that $N_k(s_0 - 1) = N'_k(s_0 - 1)$. Since $\mathcal{D}^*_\infty$ and $\mathcal{D}^{*'}_\infty$ are identical except for their $(i,k)$-th entry, by Lemma 9, we have that $N_j(s_0 - 1) = N'_j(s_0 - 1)$ for all $j$, which also implies that $\mathcal{D}_{s_0-1}$ and $\mathcal{D}'_{s_0-1}$ are identical except for the $N_k(t)$-th observation from arm $k$. Therefore, the sample mean from arm $k$ up to time $s_0 - 1$ under $\mathcal{D}'_{s_0-1}$ is larger than the one under $\mathcal{D}_{s_0-1}$.

Then, by the Exploit condition, $A_{s_0} = k$ implies that $A'_{s_0} = k$. This contradicts the assumption that $N_k(s_0) > N'_k(s_0)$. Therefore, if $X^*_{i,k} \le X^{*'}_{i,k}$ then $N_k(T)$ must be less than or equal to $N'_k(T)$. Since it holds for any $i \in \mathbb{N}$, $k \in [K]$ and $T$, the sampling strategy is optimistic, proving our claim that Exploit and IIO conditions are jointly a special case of an optimistic sampling rule. $\qquad\square$

## A.3 Sufficient conditions for Thompson sampling to be optimistic

In the previous subsection A.2, we show that Exploit and IIO conditions are jointly a sufficient condition for a sampling rule to be optimistic. In this subsection, we present a sufficient condition for Thompson sampling to satisfy both conditions, and thus to be optimistic.

For each $k$, let $\theta_k$ be the parameter of the distribution of arm $k$, and let $\mu_k = \mu(\theta_k)$. If we use an independent prior $\pi$ on $\theta := (\theta_1, \ldots, \theta_K)$, it can be easily shown that posterior distributions of $\theta$ and $\mu(\theta) := (\mu(\theta_1), \ldots, \mu(\theta_K))$ are also coordinate-wise independent conditionally on the data. Therefore, the IIO condition is trivially satisfied for the Thompson sampling algorithms. However, it is difficult to check whether the Exploit condition is satisfied because there is no closed form for $\pi(k = \mathrm{argmax}_{j \in [K]} \mu(\theta_j) | \mathcal{D}_t)$ in general.

One way to detour this issue is to study whether there exists a posterior sampling method such that the following statistically equivalent sampling algorithm satisfies the Exploit condition.

$$\nu_t(k) = \begin{cases} 1 & \text{if } k = \mathrm{argmax}_{j \in [K]} \mu_j(\theta_{j,t-1}) \\ 0 & \text{otherwise,} \end{cases}$$

where $\theta_{j,t-1}$ is a draw from the posterior distribution $\pi(\theta_j | \mathcal{D}_{t-1})$ at time $t-1$. If there exists such sampling method, we know that the sample mean from this Thompson sampling is negatively biased for any fixed $k$ and $T$. With a slight abuse of notation, we say the Thompson sampling is optimistic in this case.

For example, in Appendix A.2, we show that Thompson sampling under Gaussian arm and Gaussian prior is optimistic by using a standard Gaussian posterior sampling method described in Appendix A.1. Similarly, for the Bernoulli arm with parameters $\{p_k\}_{k=1}^K$ and beta prior with non-negative integer parameters $(n, m)$ case, we can check that the corresponding Thompson sampling is optimistic using the equivalent optimistic sampling rule

$$\nu_t(k) = \begin{cases} 1 & \text{if } k = \mathrm{argmax}_{j \in [K]} \frac{a_{j,t-1}}{a_{j,t-1} + b_{j,t-1}} \\ 0 & \text{otherwise,} \end{cases}$$

where $a_{j,t-1} = -\sum_{i=1}^{n+S_k(t-1)} \log U_{i,k}$, $b_{j,t-1} = -\sum_{i=1}^{m+N_k(t-1)-S_k(t-1)} \log W_{i,k}$ and each $U_{i,k}$ and $W_{i,k}$ are independent draws from $U(0,1)$.

In general, we have the following sufficient condition for the Thompson sampling to be optimistic.

**Corollary 10.** *Suppose the distributions of the arms belong to a one-dimensional exponential family with density $p_\eta(x) = \exp\{\eta T(x) - A(\eta)\}$ with respect to some dominating measure $\lambda$ and with*

$\eta \in$ E. *Let $\pi$ be a conjugate prior on $\eta$ with a density proportional to $\exp\{\tau\eta - n_0 A(\eta)\}$. If $\pi(\eta \leq x \mid \tau, n_0)$ is a decreasing function of $\tau$ for any given $x$ and $n_0$, and if $\eta \mapsto \mu(\eta)$ and $x \mapsto T(x)$ are both increasing or decreasing mappings, then Thompson sampling is optimistic.*

*Proof.* Fix a an arm $k \in [K]$. By the conjugacy, the posterior distribution for $\eta_k$ given the data up to time $t$ is given by

$$\pi\left(\eta_k | \mathcal{D}_t\right) \propto \exp\left\{\left(\tau + S_k^T(t)\right)\eta_k - \left(n_0 + N_k(t)\right)A(\eta_k)\right\},$$

where $S_k^T(t) := \sum_{s=1}^{t} \mathbb{1}(A_s = k)T(Y_s)$. Let $F\left(x | S_k^T(t), N_k(t)\right) := \pi\left(\eta_k \leq x | \mathcal{D}_t\right)$. From the condition on the prior, we know that $S_k^T(t) \mapsto F\left(x | S_k^T(t), N_k(t)\right)$ is a decreasing mapping for any given $x, N_k(t)$ and indices $i, k$ and $t$. Therefore $S_k^T(t) \mapsto F^{-1}\left(y | S_k^T(t), N_k(t)\right)$ is an increasing mapping for any given $y, N_k(t)$ and indices $i, k$ and $t$. Now, we can check that the Thompson sampling is equivalent to the following sampling rule.

$$\nu_t(k) = \begin{cases} 1 & \text{if } k = \text{argmax}_{j \in [K]} \mu\left(\eta_{j,t-1}\right) \\ 0 & \text{otherwise,} \end{cases}$$

where $\eta_{j,t-1} := F^{-1}\left(U_{j,t-1} | S_k^T(t-1), N_k(t-1)\right)$ and each $U_{j,t-1}$ is an independent draw from $U(0,1)$. Since $\eta \mapsto \mu(\eta)$ and $x \mapsto T(x)$ are both increasing (or decreasing), this sampling rule and the corresponding Thompson sampling is optimistic. $\square$

We can check many commonly used one-dimensional exponential family arms with its conjugate prior satisfying the condition in Corollary 10 which includes Gaussian distributions with a Gaussian prior, Bernoulli distributions with a beta prior, Poisson distributions with a gamma prior and exponential distributions with a gamma prior

### A.4 Intuitions for the sign of the bias under each optimistic sampling and stopping

Under an optimistic sampling rule with a fixed stopping time and a fixed target, Xu et al. [2013] and Nie et al. [2018] provided some intuitions as to why the sample mean is negatively biased. In this subsection, we presents a similar intuitive explanation for the negative bias of the sample mean due to adaptive sampling. We also offer some intuition in order to explain the positive bias stemming from optimistic stopping rules in the one-armed case.

For an optimistic sampling rule with a fixed stopping time, assume for simplicity that we have a fixed target arm with a symmetric distribution around its true mean. Consider two equally possible realization of the experiment up to time $t$. In one realization, the sample mean at time $t$ happens to be larger than its true mean. On the other hand, in the other scenario, the sample mean at time $t$ happens to be smaller than its true mean. In the first case, the optimistic sampling rule will draw samples more often from the target arm, and thus the sample mean will regress more easily to its true mean. In contrast, in the other case, the optimistic sampling rule will draw samples less often and thus the sample mean is less likely to regress to its true mean due to the smaller sample size. Since these two realizations are equally likely, on average, the sample mean is negatively biased. See Figure 4 for an illustration of this intuition.

For optimistic stopping in the one-armed case, consider the stopping rule that terminates the experiment when the sample mean crosses a predetermined upper boundary. See Figure 5 for an illustrative stopping boundary. As we did for the sampling case, we again assume that the distribution of the arm is symmetric around its true mean. As before, consider two equally possible realizations. In one realization, the sample mean at early times happens to be larger than the true mean. On the other hand, in the other realization, the sample means at early times is smaller than its true mean. In the first realization, the sample mean will cross the upper stopping boundary at an earlier time and thus the sample mean at the crossing time will be large. In contrast, in the other realization, the sample mean will cross the boundary at a later time and thus the optimistic stopping rule ensures that we will draw more samples in this realization and thus the sample mean is more likely to regress to its true mean due to the larger sample size. Since these two realizations are equally likely, on average, the sample mean is positively biased. See Figure 5 for an illustration of this intuition.

Figure 4: *An illustration of the intuition for why optimistic sampling results in negative bias.*

Figure 5: *An illustration of the intuition for why optimistic stopping results in positive bias.*

## B    Proofs

### B.1    Proof of Theorem 7 (the paper's central theorem on the sign of the bias)

Suppose that the data collecting strategy is monotonically decreasing for the $k$-th distribution. Then, we will first show that, for any time $t \in \mathbb{N}$, we have

$$\mathbb{E}\left[\frac{\mathbb{1}\left(\kappa = k\right)}{N_k(\mathcal{T})}\mathbb{1}\left(A_t = k\right)\left(Y_t - \mu_k\right) \mid \mathcal{F}_{t-1}\right] \leq 0. \tag{13}$$

Similarly, if the data collecting strategy is monotonically increasing, the inequality is reversed. It is understood that if $t > \mathcal{T}$, then $\mathbb{1}\left(A_t = k\right) = 0$ for all $k$, making the above claim trivially true, and hence below we implictly focus on $t \leq \mathcal{T}$.

*Proof of inequality* (13). Note that the LHS of inequality (13) can be rewritten as

$$
\begin{aligned}
&\mathbb{E}\left[\frac{\mathbb{1}\left(\kappa = k\right)}{N_k(\mathcal{T})}\mathbb{1}\left(A_t = k\right)\left(Y_t - \mu_k\right) \mid \mathcal{F}_{t-1}\right] \\
&= \mathbb{E}\left[\frac{\mathbb{1}\left(\kappa = k\right)}{N_k(\mathcal{T})}\mathbb{1}\left(A_t = k\right)\left(X_{N_k(t),k} - \mu_k\right) \mid \mathcal{F}_{t-1}\right] \\
&= \mathbb{E}\left[\sum_{i=1}^{t}\frac{\mathbb{1}\left(\kappa = k\right)}{N_k(\mathcal{T})}\mathbb{1}\left(A_t = k\right)\mathbb{1}\left(N_k(t) = i\right)\left(X_{i,k}^* - \mu_k\right) \mid \mathcal{F}_{t-1}\right] \\
&= \sum_{i=1}^{t}\mathbb{E}\left[\frac{\mathbb{1}\left(\kappa = k\right)}{N_k(\mathcal{T})}\mathbb{1}\left(A_t = k\right)\mathbb{1}\left(N_k(t) = i\right)\left(X_{i,k}^* - \mu_k\right) \mid \mathcal{F}_{t-1}\right].
\end{aligned}
$$

Therefore, it is enough to show the following inequality holds:

$$\mathbb{E}\left[\frac{\mathbb{1}\,(\kappa = k)}{N_k(\mathcal{T})}\mathbb{1}\,(A_t = k)\,\mathbb{1}\,(N_k(t) = i)\left(X_{i,k}^* - \mu_k\right)\mid \mathcal{F}_{t-1}\right] \leq 0, \qquad (14)$$

for each $t, k, i$. Recall that $\mathcal{D}_\infty^* = X_\infty^* \cup \{W_{-1}, W_0, \dots\}$ is a hypothetical dataset containing all possible independent samples from the distributions, external random sources and random seeds where the $(i, k)$-th entry of the table $X_\infty^*$ is a draw $X_{i,k}^*$ from $P_k$ independent of every other entry of $X_\infty^*$ and of the external random sources and the random seeds $\{W_{-1}, W_0, \dots\}$. Let $X_{i,k}^{*'}$ be an independent copy of $X_{i,k}^*$ and define $X_\infty^{*'}$ as a $\mathbb{N} \times K$ table that equals $X_\infty^*$ on all entries except the $(i, k)$-th entry, which contains $X_{i,k}^{*'}$. Let $\mathcal{D}_\infty^{*'} = X_\infty^{*'} \cup \{W_{-1}, W_0, \dots\}$ denote the corresponding dataset, which only differs from $\mathcal{D}_\infty^*$ in one element. Note that, for each $t$, we have

$$\mathcal{F}_t = \sigma\left(\{W_{-1}, W_0, Y_1, W_1, \dots, Y_t, W_t\}\right)$$

$$= \sigma\left(\bigcup_{k=1}^K \{X_{N_k(s),k}^*\}_{s=1}^t \cup \{W_s\}_{s=-1}^t\right),$$

because there is an one-to-one correspondence between sets of random variables generating $\sigma$-algebras. Therefore $X_{i,k}^{*'}$ is independent to $\mathcal{F}_t$ for any choice of $i, k$ and $t$.

For any $i, k$ and $t$, since $\mathbb{1}\,(A_t = k)$ and $\mathbb{1}\,(N_k(t) = i)$ are not functions of either $X_{i,k}^*$ or $X_{i,k}^{*'}$, if the data collecting strategy is monotonically decreasing, we have that

$$\mathbb{1}\,(A_t = k)\,\mathbb{1}\,(N_k(t) = i)\left(\frac{g_k(\mathcal{D}_\infty^*)}{f_k(\mathcal{D}_\infty^*)} - \frac{g_k(\mathcal{D}_\infty^{*'})}{f_k(\mathcal{D}_\infty^{*'})}\right)\left(\left(X_{i,k}^* - \mu_k\right) - \left(X_{i,k}^{*'} - \mu_k\right)\right) \leq 0.$$

Rearranging, we obtain that

$$\mathbb{1}\,(A_t = k)\,\mathbb{1}\,(N_k(t) = i)\left(\frac{g_k(\mathcal{D}_\infty^*)}{f_k(\mathcal{D}_\infty^*)}\left(X_{i,k}^* - \mu_k\right) + \frac{g_k(\mathcal{D}_\infty^{*'})}{f_k(\mathcal{D}_\infty^{*'})}\left(X_{i,k}^{*'} - \mu_k\right)\right)$$

$$\leq \mathbb{1}\,(A_t = k)\,\mathbb{1}\,(N_k(t) = i)\left(\frac{g_k(\mathcal{D}_\infty^{*'})}{f_k(\mathcal{D}_\infty^{*'})}\left(X_{i,k}^* - \mu_k\right) + \frac{g_k(\mathcal{D}_\infty^*)}{f_k(\mathcal{D}_\infty^*)}\left(X_{i,k}^{*'} - \mu_k\right)\right).$$

Next, note that $\frac{g_k(\mathcal{D}_\infty^*)}{f_k(\mathcal{D}_\infty^*)}\left(X_{i,k}^* - \mu_k\right)$ and $\frac{g_k(\mathcal{D}_\infty^{*'})}{f_k(\mathcal{D}_\infty^{*'})}\left(X_{i,k}^{*'} - \mu_k\right)$ have the same distribution and so do $\frac{g_k(\mathcal{D}_\infty^{*'})}{f_k(\mathcal{D}_\infty^{*'})}\left(X_{i,k}^* - \mu_k\right)$ and $\frac{g_k(\mathcal{D}_\infty^*)}{f_k(\mathcal{D}_\infty^*)}\left(X_{i,k}^{*'} - \mu_k\right)$. Therefore, by taking conditional expectation given $\mathcal{F}_{t-1}$ on both sides, we have

$$2\mathbb{E}\left[\mathbb{1}\,(A_t = k)\,\mathbb{1}\,(N_k(t) = i)\frac{g_k(\mathcal{D}_\infty^*)}{f_k(\mathcal{D}_\infty^*)}\left(X_{i,k}^* - \mu_k\right)\mid \mathcal{F}_{t-1}\right] \qquad (15)$$

$$\leq 2\mathbb{E}\left[\mathbb{1}\,(A_t = k)\,\mathbb{1}\,(N_k(t) = i)\frac{g_k(\mathcal{D}_\infty^*)}{f_k(\mathcal{D}_\infty^*)}\left(X_{i,k}^{*'} - \mu_k\right)\mid \mathcal{F}_{t-1}\right]$$

$$= 2\mathbb{E}\left[\mathbb{1}\,(A_t = k)\,\mathbb{1}\,(N_k(t) = i)\frac{g_k(\mathcal{D}_\infty^*)}{f_k(\mathcal{D}_\infty^*)}\mid \mathcal{F}_{t-1}\right]\mathbb{E}\left[X_{i,k}^{*'} - \mu_k\right]$$

$$= 0,$$

where the first equality comes from the fact $X_{i,k}^{*'}$ is independent of both $\frac{g_k(\mathcal{D}_\infty^*)}{f_k(\mathcal{D}_\infty^*)}$ and $\mathcal{F}_{t-1}$ and that $\mathbb{1}\,(A_t = k)$ and $\mathbb{1}\,(N_k(t) = i)$ are measurable with respect to $\mathcal{F}_{t-1}$. By plugging-in the identity $\frac{g_k(\mathcal{D}_\infty^*)}{f_k(\mathcal{D}_\infty^*)} = \frac{\mathbb{1}\,(\kappa=k)}{N_k(\mathcal{T})}$ into the LHS of (15), we obtain the inequality (14), and thus, the inequality (13) as desired. □

*Proof of the signs of the covariance and bias terms, equations* (4) *and* (6). Suppose that the data collection strategy is monotonically increasing. Consider any arm $k$ such that $\mathbb{P}(\kappa = k) > 0$.

To prove equation (4), it is enough to show that $\mathbb{E}\left[(\widehat{\mu}_\kappa - \mu_\kappa)\,\mathbb{1}(\kappa = k)\right] \leq 0$. For each $t \geq 0$, define a process that is adapted to the natural filtration $\{\mathcal{F}_t\}_{t \geq 0}$ such that $L(0) = 0$ and

$$L(t) := \mathbb{E}\left[\frac{S_k(t) - \mu_k N_k(t)}{N_k(\mathcal{T})}\mathbb{1}(\kappa = k)\mid \mathcal{F}_t\right], \quad \forall t \geq 1. \tag{16}$$

Note that the theorem requires us to show that $\mathbb{E}[L_\mathcal{T}] \leq 0$. We will first show that

$$\{L(t)\}_{t \geq 0} \text{ is a super-martingale with respect to } \{\mathcal{F}_t\}_{t \geq 0}. \tag{17}$$

First note that using inequality (13), we have

$$\mathbb{E}\left[L(1)\mid \mathcal{F}_0\right] = \mathbb{E}\left[\frac{\mathbb{1}(\kappa = k)}{N_k(\mathcal{T})}\mathbb{1}(A_1 = k)(Y_1 - \mu_k)\mid \mathcal{F}_0\right] \leq 0 = L(0).$$

Next, for all $t \geq 1$, again using inequality (13), we have

$$\mathbb{E}\left[L(t)\mid \mathcal{F}_{t-1}\right] = L(t-1) + \mathbb{E}\left[\frac{\mathbb{1}(\kappa = k)}{N_k(\mathcal{T})}\mathbb{1}(A_t = k)(Y_t - \mu_k)\mid \mathcal{F}_{t-1}\right]$$
$$\leq L(t-1).$$

(Note that since sampling stops at time $\mathcal{T}$, it is understood that for $t > \mathcal{T}$, we have $\mathbb{1}(A_t = \kappa) = 0$, $S_\kappa(t) = S_\kappa(\mathcal{T})$, $N_\kappa(t) = N_\kappa(\mathcal{T})$, $\mathcal{F}_t = \mathcal{F}_\mathcal{T}$ and thus $L(t) = L(t-1) = L(\mathcal{T})$, so the above inequality is still valid.) This proves claim (17). By the optional stopping theorem, we have that

$$\mathbb{E}L(\mathcal{T} \wedge t) \leq \mathbb{E}[L(0)] = 0, \quad \forall t \geq 1.$$

To prove $\mathbb{E}L(\mathcal{T}) \leq \mathbb{E}[L(0)]$, we follow the standard proof technique for the optional stopping theorem. To be specific, it is enough show that $|L(\mathcal{T} \wedge t)| \leq U$ for all $t \geq 0$, where $U$ is such that $\mathbb{E}[U] < \infty$. The result then follows from the dominated convergence theorem. Define $U$ as

$$U = \sum_{s=1}^{\mathcal{T}}|L(s) - L(s-1)| = \sum_{s=1}^{\infty}|L(s) - L(s-1)|\,\mathbb{1}(\mathcal{T} \geq s). \tag{18}$$

Clearly, $|L(\mathcal{T} \wedge t)| \leq U$ for all $t$. In order to show that $\mathbb{E}[U] < \infty$, first note that for any $t \geq 1$, we have

$$\mathbb{E}\left[|L(t+1) - L(t)|\mid \mathcal{F}_t\right] = \mathbb{E}\left[\frac{\mathbb{1}(\kappa = k)}{N_k(\mathcal{T})}\mathbb{1}(A_{t+1} = k)\,|Y_{t+1} - \mu_k|\mid \mathcal{F}_t\right]$$
$$\leq \mathbb{E}\left[\mathbb{1}(A_{t+1} = k)\,|Y_{t+1} - \mu_k|\mid \mathcal{F}_t\right]$$
$$= \mathbb{1}(A_{t+1} = k)\mathbb{E}\left[|Y_{t+1} - \mu_k|\mid \mathcal{F}_t\right] \tag{19}$$
$$= \mathbb{1}(A_{t+1} = k)\int |x - \mu_k|\mathrm{d}P_k(x)$$
$$:= c_k\mathbb{1}(A_{t+1} = k),$$

where the first inequality comes from the assumption $N_k(\mathcal{T}) \geq 1$ for all $k$ with $\mathbb{P}(\kappa = k) > 0$, and the following equality holds because $\mathbb{1}(A_{t+1} = k) \in \mathcal{F}_t$. The third equality stems from the observation that, on the event $(A_{t+1} = k)$, $Y_{t+1} \sim P_k$ and it is independent of the previous history. Therefore, we obtain that

$$\mathbb{E}[U] = \sum_{s=1}^{\infty}\mathbb{E}\left[\mathbb{E}\left[|L(s) - L(s-1)|\,\mathbb{1}(\mathcal{T} \geq s)\mid \mathcal{F}_{s-1}\right]\right]$$
$$= \sum_{s=1}^{\infty}\mathbb{E}\left[\mathbb{1}(\mathcal{T} \geq s)\,\mathbb{E}\left[|L(s) - L(s-1)|\mid \mathcal{F}_{s-1}\right]\right] \quad (\text{since } \mathbb{1}(\mathcal{T} \geq s) \in \mathcal{F}_{s-1}.)$$
$$\leq c_k\sum_{s=1}^{\infty}\mathbb{E}\left[\mathbb{1}(A_s = k)\,\mathbb{1}(\mathcal{T} \geq s)\right] \quad (\text{by the inequality (19)})$$
$$= c_k\mathbb{E}N_k(\mathcal{T}) < \infty,$$

where the finiteness of the last term follows from the assumption $\mathbb{E}N_k(\mathcal{T}) < \infty$ for all $k$ with $\mathbb{P}(\kappa = k) > 0$. By the dominated convergence theorem, we have that

$$\mathbb{E}\left[\widehat{\mu}_\kappa(\mathcal{T}) - \mu_\kappa\mid \kappa = k\right]\mathbb{P}(\kappa = k) = \mathbb{E}\left[(\widehat{\mu}_\kappa(\mathcal{T}) - \mu_\kappa)\,\mathbb{1}(\kappa = k)\right]$$
$$= \mathbb{E}\left[L(\mathcal{T})\right] \leq \mathbb{E}[L(0)] = 0,$$

which implies that $\mathbb{E}\left[\widehat{\mu}_\kappa \mid \kappa = k\right] \le \mu_k$. The inequality (5) follows immediately from this result and the identity

$$\mathbb{E}\left[\widehat{\mu}_\kappa(\mathcal{T}) - \mu_\kappa\right] = \sum_{k:\mathbb{P}(\kappa=k)>0} \mathbb{E}\left[\widehat{\mu}_\kappa(\mathcal{T}) - \mu_\kappa \mid \kappa = k\right]\mathbb{P}(\kappa = k).$$

Thus, the sample mean at the stopping time $\mathcal{T}$ is negatively biased.

If the data collecting strategy is monotonically increasing, the supermartingale is replaced by a submartingale and the inequalities are reversed. This observation completes the proof.

Now, suppose each arm has a bounded distribution. without loss of generality, assume there exists a fixed $M > 0$ such that $P_k\left(\left[\mu_k - M, \mu_k + M\right]\right) = 1$ for all $k \in [K]$. Then for any $t \ge 1$, we have

$$\mathbb{E}\left[|L(t+1) - L(t)| \mid \mathcal{F}_t\right] = \mathbb{E}\left[\frac{\mathbb{1}\left(\kappa = k\right)}{N_k(\mathcal{T})}\mathbb{1}\left(A_{t+1} = k\right)|Y_{t+1} - \mu_k| \mid \mathcal{F}_t\right]$$
$$\le M\mathbb{E}\left[\frac{\mathbb{1}\left(A_{t+1} = k\right)}{N_k(\mathcal{T})} \mid \mathcal{F}_t\right]. \tag{20}$$

Therefore, we obtain that

$$\mathbb{E}[U] = \sum_{s=1}^{\infty} \mathbb{E}\left[\mathbb{E}\left[|L(s) - L(s-1)|\,\mathbb{1}\left(\mathcal{T} \ge s\right) \mid \mathcal{F}_{s-1}\right]\right]$$
$$= \sum_{s=1}^{\infty} \mathbb{E}\left[\mathbb{1}\left(\mathcal{T} \ge s\right)\mathbb{E}\left[|L(s) - L(s-1)| \mid \mathcal{F}_{s-1}\right]\right] \quad (\text{since } \mathbb{1}\left(\mathcal{T} \ge s\right) \in \mathcal{F}_{s-1})$$
$$\le M\sum_{s=1}^{\infty}\mathbb{E}\left[\frac{\mathbb{1}\left(A_s = k\right)}{N_k(\mathcal{T})}\mathbb{1}\left(\mathcal{T} \ge s\right)\right] \quad (\text{by the inequality (20)})$$
$$= M < \infty \quad (\text{by the definition of } N_k(\mathcal{T})),$$

which implies that if each arm has a bounded distribution, we can determine the sign of the bias of the sample mean at the stopping time $\mathcal{T}$ without assuming $\mathbb{E}N_k(\mathcal{T}) < \infty$ for all $k$ with $\mathbb{P}(\kappa = k) > 0$.

$\square$

**About Remark 1.** In our recent work [Shin et al., 2019], we showed that if arm $k$ has a finite $p$-th moment for a fixed $p > 2$, the following bound on the normalized $\ell_2$ risk of the sample mean holds:

$$\mathbb{E}\left[\frac{N_k(\mathcal{T})}{\log N_k(\mathcal{T})}\left(\widehat{\mu}_k(\mathcal{T}) - \mu_k\right)^2\right] < \infty, \tag{21}$$

provided that $N_k(\mathcal{T}) \ge 3$. In this case, we can show that $\mathbb{E}[U] < \infty$ without assuming $\mathbb{E}N_k(\mathcal{T}) < \infty$, where $U$ is defined in (18). For each $k$, set $c_k := \int |x - \mu_k|\mathrm{d}P_k(x)$. Let $\widehat{c}_k(\mathcal{T})$ be the sample mean estimator of $c_k$ at the stopping time $\mathcal{T}$. Then, we have

$$\mathbb{E}[U] = \sum_{s=1}^{\infty} \mathbb{E}\left[\mathbb{E}\left[|L(s) - L(s-1)|\,\mathbb{1}\left(\mathcal{T} \ge s\right) \mid \mathcal{F}_{s-1}\right]\right]$$
$$= \sum_{s=1}^{\infty} \mathbb{E}\left[\frac{\mathbb{1}\left(\kappa = k\right)}{N_k(\mathcal{T})}\mathbb{1}\left(A_s = k\right)|Y_s - \mu_k|\,\mathbb{1}\left(\mathcal{T} \ge s\right)\right]$$
$$\le \mathbb{E}\left[\sum_{s=1}^{\infty}\frac{\mathbb{1}\left(A_s = k\right)}{N_k(\mathcal{T})}|Y_s - \mu_k|\,\mathbb{1}\left(\mathcal{T} \ge s\right)\right]$$
$$:= \mathbb{E}\left[\widehat{c}_k(\mathcal{T})\right]$$
$$\le \mathbb{E}\left|\widehat{c}_k(\mathcal{T}) - c_k\right| + c_k$$
$$\le \mathbb{E}\left[\sqrt{\frac{N_k(\mathcal{T})}{\log N_k(\mathcal{T})}}\left|\widehat{c}_k(\mathcal{T}) - c_k\right|\right] + c_k$$
$$\le \sqrt{\mathbb{E}\left[\frac{N_k(\mathcal{T})}{\log N_k(\mathcal{T})}\left(\widehat{c}_k(\mathcal{T}) - c_k\right)^2\right]} + c_k < \infty,$$

where in the last bound we have used (21). Thus, if each arm has a finite $p$-th moment for a fixed $p > 2$, we can determine the sign of the bias of the sample mean at the stopping time $\mathcal{T}$ without assuming $\mathbb{E}N_k(\mathcal{T}) < \infty$ for all $k$ with $\mathbb{P}(\kappa = k) > 0$.

## B.2 Proof of Corollary 8 (The lil'UCB algorithm results in positive bias)

Before presenting a formal proof of Corollary 8, we first provide an intuitive explanation why any reasonable and efficient algorithm for the best-arm identification problem would result in positive bias. For any $k \in [K]$ and $i \in \mathbb{N}$, let $\mathcal{D}_\infty^*$ and $\mathcal{D}_\infty^{*'}$ be two MAB tabular representation that agree with each other except $X_{i,k}^* < X_{i,k}^{*'}$. Since we have a larger value from arm $k$ in the second scenario $\mathcal{D}_\infty^{*'}$, if $\kappa = k$ under the first scenario $\mathcal{D}_\infty^*$, any reasonable algorithm would also pick the arm $k$ under the more favorable scenario $\mathcal{D}_\infty^{*'}$. In this case, we know that $\kappa = k$ implies $\kappa' = k$. Also note that any efficient algorithm should be able to exploit the more favorable scenario $D_\infty^{*'}$ to declare arm $k$ as the best arm by using less samples from arm $k$. Therefore, we would have $N_k(\mathcal{T}) \geq N_k'(\mathcal{T}')$. In sum, we can expect that, from any reasonable and efficient algorithm, we would have $\frac{\mathbb{1}(\kappa = k)}{N_k(\mathcal{T})} \leq \frac{\mathbb{1}(\kappa' = k)}{N_k'(\mathcal{T}')}$ which shows that the algorithm would be monotonically increasing and thus the sample mean of the chosen arm is positively biased. Below, we formally verify that this intuition works for the lil'UCB algorithm.

*Proof of Corollary 8.* For any given $i, k$, let $X_{i,k}^{*'}$ be an independent copy of $X_{i,k}^*$ and define $X_\infty^{*'}$ as a $\mathbb{N} \times K$ table which equals $X_\infty^*$ on all entries except the $(i, k)$-th entry, which contains $X_{i,k}^{*'}$. Let $\mathcal{D}_\infty^{*'} = X_\infty^{*'} \cup \{W_{-1}, W_0, \dots\}$ denote the corresponding dataset, which only differs from $\mathcal{D}_\infty^*$ in one element. Let $(N_k(T), N_k'(T))$ denote the numbers of draws from arm $k$ up to time $T$. Let $(\mathcal{T}, \mathcal{T}')$ be the stopping times and $(\kappa, \kappa')$ be choosing functions as determined by the lil'UCB algorithm under $\mathcal{D}_\infty^*$ and $\mathcal{D}_\infty^{*'}$ respectively.

Suppose $X_{i,k}^* \leq X_{i,k}^{*'}$. Proving that the lil'UCB algorithm is monotonically increasing (and hence results in positive bias) corresponds to showing that the following inequality holds:

$$\frac{\mathbb{1}(\kappa = k)}{N_k(\mathcal{T})} \leq \frac{\mathbb{1}(\kappa' = k)}{N_k'(\mathcal{T}')}. \tag{22}$$

If $\kappa \neq k$, the inequality (22) holds trivially. Therefore, for the rest of the proof, we assume $\kappa = k$ which also implies $\mathcal{T} < \infty$. (If not, the lil'UCB algorithm is not stopped, and thus $\kappa \neq k$.)

First, we can check that the lil'UCB sampling is a special case of UCB-type sampling algorithms. Therefore, it is an optimistic sampling method which implies that for any *fixed* $t > 0$, and *fixed* arm $k$, we have $N_k(t) \leq N_k'(t)$. Since $\sum_{j \neq k} N_j(t) = t - N_k(t)$ for all $t$, we can rewrite the lil'UCB stopping rule as stopping the sampling whenever there exists a $k$ such that $N_k$, which is a non-decreasing function of $t$, crosses the strictly increasing linear boundary $\left\{(n, t) : n = \frac{1 + \lambda t}{1 + \lambda}\right\}$ for a fixed $\lambda > 0$. Since $N_k(t) \leq N_k'(t)$ for all $t$, we know that $\mathcal{T}' \leq \mathcal{T}$.

Since the linear boundary is increasing, we can check $N_k'(\mathcal{T}') \leq N_k(\mathcal{T})$ if $\kappa' = k$. Therefore, to complete the proof, it is enough to show that $\kappa = k$ implies $\kappa' = k$. For the sake of deriving a contradiction, assume $\kappa = k$ but $\kappa' \neq k$. Then, there exists $j \neq k$ such that $\kappa' = j$. By the definition of $\kappa'$, it is equivalent to $N_j'(\mathcal{T}') = \max_{l \in [K]} N_l'(\mathcal{T}')$. Hence, we have that

$$N_j'(\mathcal{T}') > N_k'(\mathcal{T}'). \tag{23}$$

Similarly, we can show that

$$N_j(\mathcal{T}) < N_k(\mathcal{T}). \tag{24}$$

Since $\mathcal{T}'$ is the first time $t$ such that, for some $l$, $N_l'(t)$ has crossed the boundary, we know that $j$ is also the index of the arm which has crossed the boundary first time. Also, since the lil'UCB sampling satisfies the IIO condition, Lemma 9 along with the fact that $N_k(t) \leq N_k'(t)$ for all $t$ implies that $N_j(t) \geq N_j'(t)$ for all $j \neq k$. From the two observations above, we have the following inequalities:

$$\frac{1 + \lambda \mathcal{T}'}{1 + \lambda} \leq N_j'(\mathcal{T}') \leq N_j(\mathcal{T}'),$$

which implies that $t \mapsto N_j(t)$ is crossing the boundary at time $\mathcal{T}'$. By the definition of $\mathcal{T}$ and, by assumption, $\kappa = k$, we obtain that $\mathcal{T} \leq \mathcal{T}'$.

Similarly, from the fact that $N_k(t) \leq N_k'(t)$ for all $t$ along with the definition of $\mathcal{T}$, we have that

$$\frac{1 + \lambda \mathcal{T}}{1 + \lambda} \leq N_k(\mathcal{T}) \leq N_k'(\mathcal{T}),$$

which implies that $t \mapsto N_k'(t)$ is crossing the boundary at time $\mathcal{T}$, and thus $\mathcal{T}' \leq \mathcal{T}$ since $\kappa' \neq k$ by assumption.

From the two observations above, we have $\mathcal{T}' = \mathcal{T}$. Finally, note that

$$N_k'(\mathcal{T}') < N_j'(\mathcal{T}') \leq N_j(\mathcal{T}') = N_j(\mathcal{T}) < N_k(\mathcal{T}) \leq N_k'(\mathcal{T}) = N_k'(\mathcal{T}')$$

where the first inequality comes from the inequality (23). The second inequality come from $N_j' \leq N_j$. The first equality comes from $\mathcal{T}' = \mathcal{T}$ and the third inequality comes from the inequality (24). The last inequality comes from $N_k \leq N_k'$ and the final equality comes from $\mathcal{T} = \mathcal{T}'$.

This is a contradiction, and, therefore, $\kappa = k$ implies that $\kappa' = k$. This proves that the lil'UCB algorithm is monotonically increasing and the chosen stopped sample mean from the lil'UCB algorithm is positively biased. $\qquad\square$

### B.3  Proof of Proposition 5 (bias expression) via Lemma 6 (Wald's identity for MAB)

By direct substitution, we first note that

$$\mathbb{E}\left|S_k(\mathcal{T}) - \mu_k N_k(\mathcal{T})\right| = \mathbb{E}\left[\sum_{t=1}^{\infty} \mathbb{1}\left(A_t = k\right)\left|Y_t - \mu_k\right|\mathbb{1}\left(\mathcal{T} \geq t\right)\right]$$

$$= \sum_{t=1}^{\infty} \mathbb{E}\left[\mathbb{1}\left(A_t = k\right)\left|Y_t - \mu_k\right|\mathbb{1}\left(\mathcal{T} \geq t\right)\right]$$

$$= \sum_{t=1}^{\infty} \mathbb{E}\left[\mathbb{1}\left(A_t = k\right)\mathbb{1}\left(\mathcal{T} \geq t\right)\mathbb{E}\left[\left|Y_t - \mu_k\right| \mid \mathcal{F}_{t-1}\right]\right]$$

$$= \sum_{t=1}^{\infty} \mathbb{E}\left[\mathbb{1}\left(A_t = k\right)\mathbb{1}\left(\mathcal{T} \geq t\right)\int |x - \mu_k| \mathrm{d}P_k(x)\right]$$

$$= \int |x - \mu_k| \mathrm{d}P_k(x)\mathbb{E}\left[\sum_{t=1}^{\infty} \mathbb{1}\left(A_t = k\right)\mathbb{1}\left(\mathcal{T} \geq t\right)\right]$$

$$= \int |x - \mu_k| \mathrm{d}P_k(x)\mathbb{E}\left[N_k(\mathcal{T})\right] < \infty,$$

where the second equality comes from the Tonelli's theorem and the third equality stems from the facts that $\mathbb{1}(A_t = k)$ and $\mathbb{1}(\mathcal{T} \geq t)$ are $\mathcal{F}_{t-1}$ measurable. The fourth equality comes from the fact that, on event $\mathbb{1}(A_t = k)$, $Y_t \sim P_k$ and it is independent of the previous history. Finally, the finiteness of the last term comes from the assumption of the existence of the first moment of $k$-th arm and $\mathbb{E}[N_k(\mathcal{T})] < \infty$. Therefore, by the dominated convergence theorem, we have

$$\mathbb{E}\left[S_k(\mathcal{T}) - \mu_k N_k(\mathcal{T})\right] = \mathbb{E}\left[\sum_{t=1}^{\infty} \mathbb{1}\left(A_t = k\right)\left[Y_t - \mu_k\right]\mathbb{1}\left(\mathcal{T} \geq t\right)\right]$$

$$= \sum_{t=1}^{\infty} \mathbb{E}\left[\mathbb{1}\left(A_t = k\right)\left[Y_t - \mu_k\right]\mathbb{1}\left(\mathcal{T} \geq t\right)\right]$$

$$= \sum_{t=1}^{\infty} \mathbb{E}\left[\mathbb{1}\left(A_t = k\right)\mathbb{1}\left(\mathcal{T} \geq t\right)\mathbb{E}\left[Y_t - \mu_k \mid \mathcal{F}_{t-1}\right]\right]$$

$$= 0,$$

which implies $\mu_k \mathbb{E}\left[N_k(\mathcal{T})\right] = \mathbb{E}\left[S_k(\mathcal{T})\right]$, which proves the generalization of Wald's first identity.

Since $\mathbb{E}[N_k(\mathcal{T})] > 0$, one can then express $\mu_k$ as

$$\mu_k = \frac{\mathbb{E}[S_k(\mathcal{T})]}{\mathbb{E}[N_k(\mathcal{T})]}.$$

By direct substitution, the bias of the sample mean can thus be expressed as

$$\begin{aligned}
\mathbb{E}[\widehat{\mu}_k(\mathcal{T}) - \mu_k] &= \mathbb{E}\left[\widehat{\mu}_k(\mathcal{T})\left(1 - \frac{N_k(\mathcal{T})}{\mathbb{E}[N_k(\mathcal{T})]}\right)\right] \\
&= \mathrm{Cov}\left(\widehat{\mu}_k(\mathcal{T}), \left(1 - \frac{N_k(\mathcal{T})}{\mathbb{E}[N_k(\mathcal{T})]}\right)\right) \\
&= -\frac{\mathrm{Cov}\left(\widehat{\mu}_k(\mathcal{T}), N_k(\mathcal{T})\right)}{\mathbb{E}[N_k(\mathcal{T})]}.
\end{aligned}$$

This completes the proof of the proposition.

## C   Additional simulation results

### C.1   More on negative bias due to optimistic sampling

We conduct a simulation study in which we have three unit-variance Gaussian arms with $\mu_1 = 1, \mu_2 = 2$ and $\mu_3 = 3$. After sampling once from each arm, greedy, UCB and Thompson sampling are used to continue sampling until $T = 200$. We repeat the whole process from scratch $10^4$ times for each algorithm to get an accurate estimate for the bias.

For UCB, we use $u_{t-1}(s, n) = \sqrt{\frac{2\log(1/\delta)}{n}}$ with $\delta = 0.1$. For Thompson sampling, we use independent standard Normal priors for simplicity. We repeat the whole process from scratch 2000 times for each algorithm to get an accurate estimate for the bias.

Figure 6 shows the distribution of observed differences between sample means and the true mean for each arm under the greedy algorithm. Vertical lines correspond to biases. The example demonstrates that the sample mean is negatively biased under optimistic sampling rules. Similar results from UCB / Thompson sampling algorithms can be found in Section 4.1.

Figure 6: *Data is collected by the greedy algorithm from three unit-variance Gaussian arms with $\mu_1 = 1, \mu_2 = 2$ and $\mu_3 = 3$. For all three arms, sample means are negatively biased.*

## C.2  Positive bias from optimistic choosing and stopping in identifying the largest mean

Suppose we have $K$ arms with mean $\mu_1, \ldots, \mu_K$. As we were in Section 4.3, we are interested not in each individual arm but in the arm with the largest mean. That is, our target of inference is $\mu_* := \max_{k \in [K]} \mu_k$.

Instead of using the lil'UCB algorithm, we can draw a sample from each arm in a cyclic order for each time $t$ and use a naive sequential procedure based on the following stopping time.

$$\mathcal{T}_M^\delta := \inf \left\{ t \in \{K, 2K, \ldots, MK\} : \widehat{\mu}_{(1)}(t) > \widehat{\mu}_{(2)}(t) + \delta \right\}, \tag{25}$$

where $M, \delta > 0$ are prespecified constants and $\widehat{\mu}_{(k)}(t)$ is the $k$-th largest sample mean at time $t$. Once we stop sampling at time $\mathcal{T}_M^\delta$, we can estimate the largest mean by the largest stopped sample mean $\widehat{\mu}_{(1)}\left(\mathcal{T}_M^\delta\right)$.

The performance of this sequential procedure can vary based on underlying distribution of the arm and the choice of $\delta$ and $M$. However, we can check this optimistic choosing and stopping rules are jointly monotonic increasing and thus the largest stopped sample mean $\widehat{\mu}_{(1)}\left(\mathcal{T}_M^\delta\right)$ is always positively based for any choice of $\delta$ and $M$.

To verify it with a simulation, we set 3 unit-variance Gaussian arms with means $(\mu_1, \mu_2, \mu_3) = (g, 0, -g)$ for each gap parameter $g = 1, 3, 5$. We conduct $10^4$ trials of this sequential procedure with $M = 1000$ and $\delta = 0.7 \times g$. Figure 7 shows the distribution of observed differences between the chosen sample means and the corresponding true mean for each $\delta$. Vertical lines correspond to biases. The simulation study demonstrate that, in all configurations, the largest stopped sample mean $\widehat{\mu}_{(1)}\left(\mathcal{T}_M^\delta\right)$ is always positively biased. Note, in contrast to the lil'UCB case in Section 4.3, we have a larger bias for a smaller gap since the number of sample sizes are similar for each gaps due to the adaptive (and oracle) choice of the parameter $\delta$ but a smaller gap makes more difficult to identify largest mean correctly.

Figure 7: *Data is collected by the sequential procedure described in Appendix C.2 under unit-variance Gaussian arms with $\mu_1 = g, \mu_2 = 0$ and $\mu_3 = -g$ for each gap parameter $g = \{1, 3, 5\}$. For each gap $g$, we set the parameter $\delta = 0.7 \times g$ and $M = 1000$. For all cases, chosen sample means are positively biased.*



[Supplementary Material 2]



Estimate largest mean
Bias = (0.001, 0.002, 0.006)

Case
Small gap (g=1)
Medium gap (g=3)
Large gap (g=5)

density

difference between sample and true means

Estimate largest mean
Bias = (0.001, 0.002, 0.006)

density

Case

Small gap (g=1)
Medium gap (g=3)
Large gap (g=5)

difference between sample and true means

[Supplementary Material 3]



**Estimate largest mean**
Bias = (0.001, 0.002, 0.006)

**Estimate largest mean**
Bias = (0.001, 0.002, 0.006)

Case
- Small gap (g=1)
- Medium gap (g=3)
- Large gap (g=5)

density

difference between sample and true means

[Supplementary Material 4]



**Under the null hypothesis**
Bias = (0.01, −0.01)

**Under the alternative hypothesis**
Bias = (0.091, −0.091)

Arms
- mu1 = 0
- mu2 = 0

Arms
- mu1 = 1
- mu2 = 0

density

difference between sample and true means

[Supplementary Material 5]



**UCB algorithm**
Bias = (−0.291, −0.363, −0.006)

**Thompson sampling**
Bias = (−0.081, −0.262, −0.106)

Arms
- N(1,1)
- N(2,1)
- N(3,1)

density

difference between sample and true means