[Reviews · NeurIPS 2019]

Reviewer 1



===== Summary ===== This paper studies the bias in the sample mean estimates of actions in a multi-armed bandit setting. More specifically, the authors consider three sources of bias: the adaptive sampling of the arms, the adaptive stopping time, and the adaptive best-arm identification (choosing) strategy. It generalizes and complements the results of Nie et al. (AISTATS 2018). ===== Strengths ===== [Originality] * The authors provide a different view (to my knowledge) of the bandit setting (the tabular setting). * The contributions build on previous work (Nie et al. 2018), but are more general and do bring something new. [Quality] * The authors provide several examples and numerical experiments to illustrate/support their claims. * I didn't review appendix proofs in detail, but they seemed sound from an overview. [Clarity] * The paper is well organized and reasonably easy to follow given the sometimes uncommon notation. [Significance] * If bandits (and RL algorithms in general) are to be deployed more and more (e.g. in adaptive trials, A/B testing), I think that it is essential to understand better the potential bias in the resulting action estimates. ===== Weaknesses ===== [Clarity] * The writing could be polished. [Significance] * The contributions could have a broader impact if they were better linked to examples of applications where it is relevant to know the sign of bias. At first sight, it is not clear how learning about the sign of bias, but not the amplitude of bias, can be useful. Questions: * Is the choosing strategy always adaptive? If not, can you provide an example of non-adaptive choosing? Minor details: * Some results (e.g. Fig. 2-left and Fig. 3) are difficult to see. Maybe these could be zoomed to really see on which side fall the estimates? * The term "optimism" is already well-known and used in bandits for something else -- it may confuse the reader. * Bad cross-references need to be fixed in the introduction. * Sec. 2.1: Explicitly define A_t and Y_t. * The title could be improved. -------------------------------------------------------------------- I have read the rebuttal -- this is a good paper. Now that you mention it, the title is a nice reply to Nie et al. Linking explicitly the term "optimism" to the "optimism in front of uncertainty" principle should indeed help the reader, in addition to situating the paper w.r.t. the bandits literature (as pointed out by reviewer 2). Non-adaptive choosing strategy: I agree that they may exist in theory, but they are counterintuitive when speaking about bandits, which typically arise from adaptive experimental designs, therefore aiming at being adaptive by definition. It may be worth mentioning.

Reviewer 2



TL;DR: I learnt something by reading this paper and I think a lot of people working on bandits and adaptive algorithms should be aware of those results. I am also happy with how they corrected/completed the Nie et al. 2018 paper that was quite unclear to me. Originality : good. it is an unusual read in this domain (especially when one hasn't read the paper by Nie et al. 2018) Quality : Very good. It is a technically sound paper that defines and solves a precise result; Clarity : good. It is well-written but I would highly recommend to add references to the bandit literature as it is this is the targeted community; I must admit I was a bit lost at the beginning, I wasn't sure we were talking about the same thing. Significance: Okay. I think this is possibly the weakest point because in the end, I don't know exactly which issue(s) this result will solve... Major comments: M1. This paper could be of interest to a lot of people in the bandit community which is large and diverse... but for them to know about it, you have to cite them ! This paper has a total of 10 citations, including the paper they correct and a bunch of math papers that look awesome... but nearly no Bandit paper ! As far as I'm concerned, it made the paper pretty abstract as I felt it was not solving any of the problems I care about. And I still think it doesn't really, but it does bring some understanding about all those problems. I would definitely recommend expliciting further the links with the literature and I suggest at the end of this review a list of possible references to add. Typically, for the average bandit person, it feels weird to see UCB / TS / lil'UCB all mixed together and compared because they target really different kinds of problems, namely Best Arm Identification versus Cumulative Regret Minimization. I understand the scope of the paper is wider than those distinctions but that could be made clear somewhere. M2. So regarding the question of which problem is solved by this paper, I am a bit puzzled. I get that in Nie et al 2018, the message is that there analysis of the bias would allow a practitioner to correct the sample means obtained at the stopping time so that they return a less biased value. But it is not made very clear in your paper that this could be a use case of Theorem 7. Do you think you could make the goal of this analysis a bit more explicit ? M3. Numerical experiments. To be a fair comparison between UCB and TS, you should use the anytime confidence bonus \sqrt{\frac{2\log(t)}{N_k(t)}} cf KL-UCB-switch: optimal regret bounds for stochastic bandits from both a distribution-dependent and a distribution-free viewpoints (https://arxiv.org/abs/1805.05071) and the MOSS paper. I mean, otherwise it's hard to explain why you get biases an order of magnitude bigger for UCB than TS. I am not sure if this "fair comparison" would bring more answers or more questions. A general interrogation I have by reading your paper is: Could we get a relationship between the performance of a bandit algorithm (i.e it's regret in general) and the bias at some stopping time ? In the regret minimization setting, a reasonable possible relationship would be: The smaller the regret, the bigger the bias at horizon T, because smaller regret means less exploration and so less data. Similar analysis could be done on the BAI problem. Those are just suggestions for further developments rather than real critics. minor comments : m1: Section refs on p2 broken Refs : The Bandit Algorithms book by Lattimore and Szepesvari: contains literally everyting about the bandit literature (if only one, at least this one) @book{lattimore2019book, title = {Bandit Algorithms}, author = {Lattimore, Tor and Szepesv\'{a}ri, Csaba}, year = {2019}, publisher = {Cambridge University Press (preprint)}, } Optimal best arm identification with fixed confidence: a different type of Best Arm Identification @inproceedings{garivier2016optimal, title={Optimal best arm identification with fixed confidence}, author={Garivier, Aur{\'e}lien and Kaufmann, Emilie}, booktitle={Conference on Learning Theory}, pages={998--1027}, year={2016} } The original Auer 2002 paper that coined the term "optimism principle" as far as I know: @article{auer2002using, title={Using confidence bounds for exploitation-exploration trade-offs}, author={Auer, Peter}, journal={Journal of Machine Learning Research}, volume={3}, number={Nov}, pages={397--422}, year={2002} } A very similar effect noticed in management science and decision analysis : @article{smith2006optimizer, title={The optimizer’s curse: Skepticism and postdecision surprise in decision analysis}, author={Smith, James E and Winkler, Robert L}, journal={Management Science}, volume={52}, number={3}, pages={311--322}, year={2006}, publisher={INFORMS} } ---------------------------------------------------------------------------------------- Edit after rebuttal : We thank the authors for their answers to my and other reviewers' questions. We hope the NeurIPS community will get the occasion to learn as much as we had and we look forward to future developments around this idea.

Reviewer 3



Quality : All the checked calculations looked good to me. Significance and Originality : As mentioned above, this paper provides a powerful tool to determine the sign of the bias in sequential sampling. While it is "folklore intuition" that UCB and co. have negative bias, to my knowledge, this rigorous proof is new (with inspiration from the cited papers) and should pique the curiosity of anyone involved with bandits. Furthermore, this subsumes this intuition as they also consider random stopping and choosing rules. Clarity : Apart from some awkwardly chosen terminology (e.g., optimism, monotonically increasing), the paper was overall very well written and structured, with some welcome examples. Post-rebuttal edit : Nothing to add

[Author Response · NeurIPS 2019]

We thank the reviewers for their careful reading of the manuscript and for their comments.

**Response to reviewer #1** Thank you for your positive reception. We will polish the writing and address the minor comments (cross-references, readability of Fig. 2,3, etc). We are happy to consider any suggestion you may have for an altered title in the final review (that may arise in the discussion): ours was meant as a rebuttal to the Nie et al. title.

[...*potential applications where the sign of bias can be useful...*] The sign of the bias has been used to check whether a search advertising system charged advertisers fairly [Xu et al., 2013] or to investigate the validity of MAB models for the design of clinical trials [Villar et al., 2015]. We will add these examples to the paper. As the reviewer points out, quantifying the magnitude of bias is also an important complementary problem, which we will pursue in future work.

[...*the term "optimism" is well-known and used in bandits...*] Our choice of the term optimism is deliberate, and is directly inspired by the principle of "optimism in the face of uncertainty" that is frequently invoked in the MAB literature. Indeed, we view our definitions of optimistic sampling, stopping and choosing as different ways of formalizing that very same principle. (For example, UCB algorithms are optimistic samplers, and our definition captures this and other methods.) We will include an explicit reference to the principle of optimism and clarify our choice of terminology. We thought hard about a different term (and looked up synonyms), but could not find a more appropriate one, and are happy to listen to a suggestion in the final review.

[...*Is the choosing strategy always adaptive?...*] The choosing strategy can be non-adaptive. For instance, choosing a target arm by using an independent uniform distribution on $\{1, \ldots, K\}$, is a random but non-adaptive choice.

**Response to reviewer #2**

[M1. *This paper could be of interest to people in the bandit community...you have to cite them more.*] We agree with the reviewer that it would be appropriate and beneficial to the manuscript to include more references to the MAB literature and we will do so, starting from the reviewer's suggestions.

[M2. *...make the goal of this analysis more explicit.*] The main objective of this paper is to provide a rigorous and comprehensive analysis of the sign of the bias in MAB settings under adaptive sampling, stopping and choosing rules. This is, we believe, a problem of practical and theoretical significance that has not been fully solved. For instance, consider the offline analysis of data that was collected by an MAB algorithm (with any aim): suppose that you want to estimate the mean reward of some of the better arms that were picked more frequently by the algorithm. The paper by Nie et al. proves that the the sample mean will be negatively biased under fairly common adaptive sampling rules: this could instill a possibly false sense of comfort with your sample mean estimate since their theory suggests that you are underestimating the effect size. However, we prove that if the algorithm was adaptively stopped and the arm index was adaptively picked, then the net bias can actually be positive. Indeed, we prove that this is the case for lil'UCB, but it is likely true more generally as captured by our main theorem. Thus, the sample mean may actually overestimate the effect size. This is an important and general phenomenon for both theoreticians (to study further and quantify) and for practitioners (to pay heed to) because if a particular arm is later deployed in practice, it may yield a lower reward than was possibly expected from the offline analysis. We can add such an explanation to the paper if the reviewer found it useful to compare messages explicitly.

[M3. *...relationship between the performance of a bandit algorithm and the bias?...fair comparison between UCB and TS in numerical experiments...*] This is a great question/suggestion, which we can study in future work. The main goal in our simulations is to visualize and corroborate our theoretical results about the sign of the bias. As a result, we did not make any attempt to optimize the parameters for UCB or TS for the purpose of minimizing the regret, since the latter was not the paper's aim. We will include a remark in the paper to clarify this.

**Response to reviewer #3**

Thank you for your detailed review which captures the most important features of our work concisely.

# References

Sofía S Villar, Jack Bowden, and James Wason. Multi-armed bandit models for the optimal design of clinical trials: benefits and challenges. *Statistical science: a review journal of the Institute of Mathematical Statistics*, 30(2):199, 2015.

Min Xu, Tao Qin, and Tie-Yan Liu. Estimation bias in multi-armed bandit algorithms for search advertising. In *Advances in Neural Information Processing Systems*, pages 2400–2408, 2013.


[Meta-Review · NeurIPS 2019]

Although the results presented in this paper are probably of moderate impact, the quality and clarity of this work is highlighted by all three reviewers. I thus defend acceptance. I would just recommend to implement the suggestions they make for the final version.